# STARCaster: Spatio-Temporal AutoRegressive Video Diffusion for Identity- and View-Aware Talking Portraits

**Foivos Paraperas Papantoniou** [1]   **Stathis Galanakis** [1]   **Rolandos Alexandros Potamias** [1]   **Bernhard Kainz** [1 2]
**Stefanos Zafeiriou** [1]

https://foivospar.github.io/STARCaster/

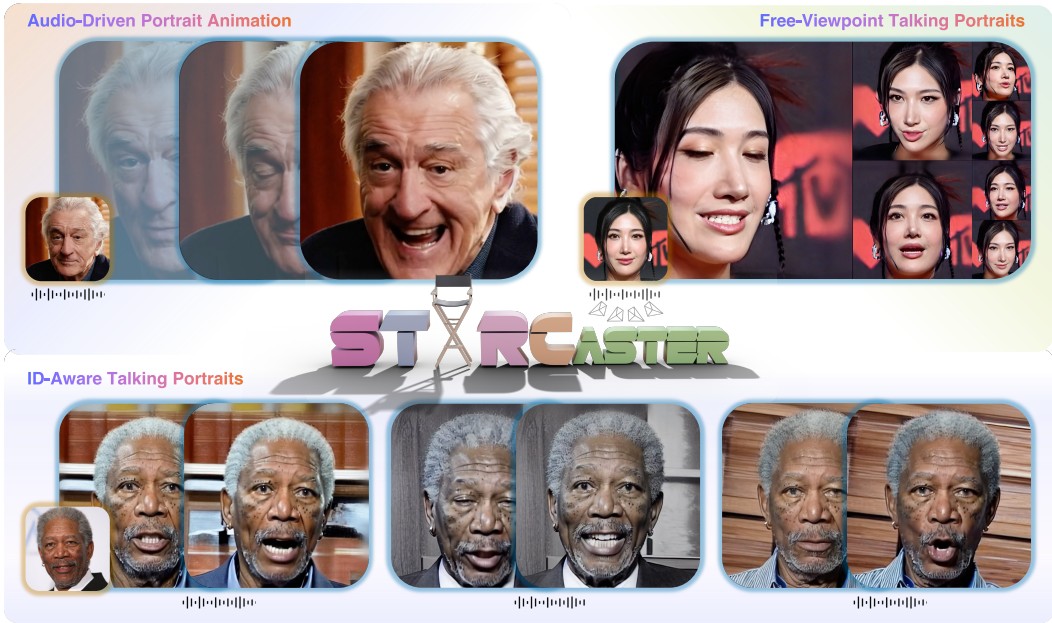

*Figure 1.* STARCaster is a Spatio-Temporal AutoRegressive model that unifies speech-driven portrait animation and continuous view synthesis in a single video diffusion framework, without relying on explicit 3D representations. Leveraging strong ID guidance, it further supports subject-consistent yet reference-free talking portraits, allowing recontextualization beyond the constraints of the input image.

## Abstract

This paper presents STARCaster, an identity-aware spatio-temporal video diffusion model that addresses both speech-driven portrait animation and dynamic viewpoint control, given an identity embedding or reference image, within a unified framework. Existing 2D speech-to-video diffusion models depend heavily on reference guidance, leading to limited motion diversity. At the same time, 3D-aware animation typically relies on inversion through pretrained tri-plane generators, which often leads to imperfect reconstructions and identity drift. We rethink reference- and geometry-based paradigms in two ways. First, we deviate from strict reference conditioning at pretraining by introducing softer identity constraints. Second, we address 3D awareness implicitly within the 2D video domain by leveraging the inherent multi-view nature of video data. STARCaster adopts a compositional approach progressing from ID-aware motion modeling, to audio-visual synchronization via lip reading-based supervision, and finally to novel view animation through temporal-to-spatial adaptation. To overcome the scarcity of 4D audio-visual data, we propose a decoupled learning approach in which view consistency and temporal coherence are trained independently. Comprehensive evaluations demonstrate that STARCaster generalizes effectively across tasks and identities, consistently surpassing prior approaches in different benchmarks.

[1]Imperial College London, UK [2]FAU Erlangen–Nürnberg, Germany. Correspondence to: Foivos Paraperas Papantoniou <f.paraperas@imperial.ac.uk>.

*Proceedings of the $43^{rd}$ International Conference on Machine Learning*, Seoul, South Korea. PMLR 306, 2026. Copyright 2026 by the author(s).

# 1. Introduction

With the rise of generative AI in the video domain, one of the most popular applications is audio-driven human animation, aiming to animate a person's image in accordance with a speech signal. Such technology holds significant potential across various industries, from filmmaking and AI agents to telepresence and virtual communication. Yet, despite vast literature, most talking portrait methods primarily emphasize technical challenges, such as audio-visual synchronization or long-term generation, often overlooking crucial aspects such as view control or diversity of generation, capabilities that have already been demonstrated in 2D human synthesis with text-to-image models.

Early approaches to portrait image animation relied heavily on GAN-based frameworks (Prajwal et al., 2020; Zhang et al., 2023), typically employing feature warping or motion transfer to achieve one-shot animation of unseen subjects. While they enabled plausible facial motion, they suffered from limited realism due to the scarcity of high-quality training data and the fundamental scaling challenges of GANs (Goodfellow et al., 2014). More recently, the field has shifted towards diffusion animation models (Tian et al., 2024; Xu et al., 2024a; Chen et al., 2025), where stronger generative priors, large-scale data, and increased resources have enabled significant improvements in fidelity and temporal coherence. In parallel, research in 3D modeling has facilitated human avatar reconstruction and animation. Building on efficient representations, such as Neural Radiance Fields (NeRFs) (Mildenhall et al., 2021) and Gaussian Splatting (Kerbl et al., 2023), many methods attempt to recover a 3D representation of a subject from 2D input data. Currently, most successful approaches typically rely on person-specific supervision, such as short driving videos, multi-view image captures (Zielonka et al., 2023; Qian et al., 2024), or, in some cases, even single images distilled through 2D diffusion priors (Gerogiannis et al., 2025). However, they are limited by test-time optimization. Instead, the remarkable generalization of foundation diffusion models (Ho et al., 2020) has opened the possibility of training-free view synthesis, with early works demonstrating compelling results for objects and scenes (Shi et al., 2023; Gao* et al., 2024). Yet, crafting realistic and rotatable one-shot talking portraits remains challenging. Current solutions (Li et al., 2023b; Ye et al., 2024) largely depend on explicit 3D priors to map a 2D input into a reconstructed 3D space, rather than directly learning spatio-temporal consistency in the video domain.

In this work, we present a 2D autoregressive video diffusion model for audio-driven face animation that simultaneously generalizes to novel views, without requiring explicit 3D modeling. We build upon a pretrained ID-aware image diffusion backbone (Paraperas Papantoniou et al., 2024), extending it to video generation through lightweight archi-tectural adaptations. Using a careful conditioning mechanism, we retain its strong identity prior, enabling the generation of diverse yet ID-consistent talking portraits, as illustrated in Figure 1 and Figure 2. To learn novel-view animation, we employ a dataset of synthetic 3D heads to derive consistent spatial trajectories for training, effectively reformulating view control as a video generation task. Our framework explicitly leverages spatio-temporal consistency in videos, allowing for both speech-driven motion and continuous viewpoint manipulation at test time. Additionally, we incorporate lip-reading supervision and a self-forcing-based training strategy to enhance lip synchronization accuracy and motion expressiveness beyond existing approaches. Overall, we make the following contributions:

- We propose a unified spatio-temporal autoregressive diffusion model for both one-shot audio-driven portrait animation and view synthesis. Beyond explicit portraits as input, our model leverages identity embeddings to generate diverse animations of input subjects.

- We demonstrate that combining large-scale video-based spatial priors with limited synthetic multi-view data achieves excellent cross-view coherence without relying on explicit 3D representations.

- We introduce a self-forcing-based training scheme that helps overcome static facial animations. Our experiments demonstrate superior performance to prior approaches in multiple tasks.

# 2. Related Work

**Diffusion Models for Video Generation.** Building on the success of text-to-image models, recent years have witnessed remarkable progress in video diffusion models (Ho et al., 2022). Initial efforts, including Stable Video Diffusion (Blattmann et al., 2023a), Make-A-Video (Singer et al., 2022), MagicVideo (Zhou et al., 2022), Align Your Latents (Blattmann et al., 2023b), and AnimateDiff (Guo et al., 2024), primarily adopted pre-existing 2D UNet architectures, extended with temporal attention modules to enable text-to-video generation by fine-tuning on curated video datasets. Subsequent approaches aimed at improving controllability and motion modeling. For instance, Video-Composer (Wang et al., 2023) incorporates spatial, temporal, and textual conditions, while SparseCtrl (Guo et al., 2023) provides structure control from sparse signals such as sketches, depth maps, or RGB keyframes. Several works (Xu et al., 2024b; Hu et al., 2023; Zhu et al., 2024; Wang et al., 2024b; Zhang et al., 2025; Ma et al., 2023a; Chang et al., 2025) have further adapted this paradigm to human-centric generation, employing 3D body models, skeleton sequences, or dense-pose representations to guide animation.

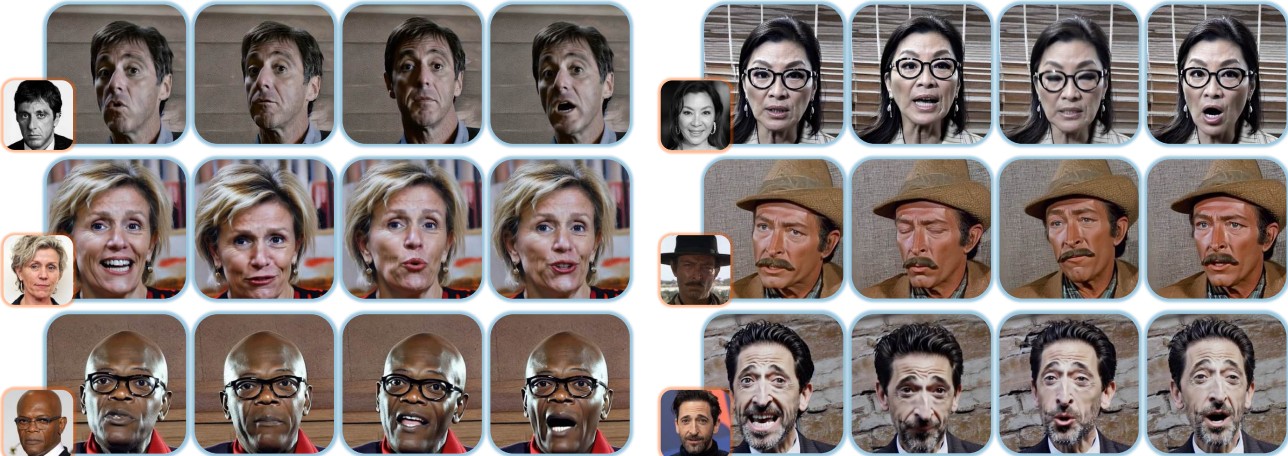

*Figure 2.* Novel talking instances of arbitrary subjects generated by STARCaster, conditioned on identity features and a driving audio.

Recently, attention has shifted toward scaling video models for world modeling via billion-parameter Diffusion Transformers (DiTs) (Peebles & Xie, 2023). Frontier systems (Polyak et al., 2024; Kong et al., 2024; Wan et al., 2025; Yang et al., 2024) are typically developed within industry, where extensive resources and large proprietary datasets are available. However, such models remain computationally intensive, requiring multi-GPU inference and substantial memory footprints, which limits their suitability for interactive applications.

**Speech-driven Portrait Animation.** The task of one-shot audio-driven animation, where the model must generalize to unseen identities, was first addressed by a series of GAN-based methods. Wav2Lip (Prajwal et al., 2020) was among the first to achieve accurate lip synchronization using a pretrained expert lip-sync discriminator, while SadTalker (Zhang et al., 2023) introduced an audio-to-expression module predicting motion coefficients followed by a 3D-aware GAN-based face renderer. AniTalker (Liu et al., 2024b) further proposed a universal motion representation to handle complex facial dynamics. Departing from frame-by-frame generation, recent works have adopted video diffusion to generate sequences collectively with greater temporal coherence. In this context, several works (Shen et al., 2023; Wei et al., 2024; Tian et al., 2024; Xu et al., 2024a; Cui et al., 2024; 2025; Jiang et al., 2024; Chen et al., 2025; Wang et al., 2024a; Ki et al., 2025) utilize audio features, sometimes augmented with landmarks or other structural cues, to guide video synthesis with attention-based conditioning mechanisms. Due to the high computational cost and slow inference of video diffusion, most existing approaches adopt autoregressive inference to extend animation length (Xu et al., 2024a; Jiang et al., 2024; Tian et al., 2024). Such models typically generate fixed-length segments at a time, conditioned on the reference frame and previously synthesized frames. While this strategy enables

longer generation, it often produces overly static sequences with limited motion, as appearance is largely copied from the reference portrait. Our approach deviates in three ways: (1) We inherit strong subject consistency through a prior ID-conditioned image expert. (2) Before the reference animation task, we pretrain the video model with ID and audio conditioning only, enabling free motion synthesis without reliance on repeated reference-frame conditioning. (3) We implicitly expose the model to longer temporal contexts through autoregressive self-forcing-based training. Together, these steps yield higher ID similarity and motion diversity, reducing the "copy-paste" effects typical of existing autoregressive methods.

**One-Shot 3D-Aware Talking Portraits.** Reconstructing animatable 3D face avatars has also been studied extensively, with methods adopting a variety of representations ranging from meshes (Grassal et al., 2022; Gerogiannis et al., 2024) to neural fields (Gafni et al., 2021; Athar et al., 2022; Zheng et al., 2022; Zielonka et al., 2023; Zheng et al., 2023) and, more recently, Gaussian Splats (Qian et al., 2024). However, they typically require multi-view or video data. Despite efforts toward single-image reconstruction using diffusion priors (Babiloni et al., 2024; Gerogiannis et al., 2025; Taubner et al., 2025), such approaches still require per-subject optimization. The task of one-shot, 3D-aware talking portrait animation is even more challenging, with few methods addressing it directly. OTAvatar (Ma et al., 2023b) introduces a motion controller to decouple identity and motion in the latent space, coupled with a tri-plane generator, while (Li et al., 2023a) learns an image encoder to predict canonical volume features and an expression-aware deformation module to drive the 3D model. Li et al. (Li et al., 2023b) propose a three-branch framework for geometry, appearance, and expression, followed by volumetric rendering and super-resolution. Similarly, Real3DPortrait (Ye et al., 2024) adopts a large image-to-plane architecture enhancing 3D

animation of "in-the-wild" portraits, whereas (Deng et al., 2024) converts monocular videos into pseudo multi-view representations, which are then used to train an image-to-4D head synthesizer. IM-Portrait (Li et al., 2025) instead employs a diffusion model to generate Multiplane Images for 4D talking head synthesis. Current approaches mainly focus on mapping input images to animatable 3D representations, such as tri-planes, which can introduce fitting errors and artifacts. Moreover, few natively support audio-driven animation without explicit driving expressions (Ye et al., 2024). In contrast, we leverage a powerful video model that achieves implicit 3D-aware generalization through lightweight fine-tuning on pseudo multi-view data, avoiding reliance on imperfect 3D reconstruction.

# 3. Method

We propose a video diffusion model derived from an identity-conditioned backbone, which further integrates audio, viewpoint and portrait conditioning in a disentangled manner through a multi-source attention mechanism and a reference network. We also adopt a progressive training strategy that advances from identity-aware audio-visual motion modeling to long-term talking video generation and finally to spatial generation for view synthesis. Below, we detail the key components of our approach. A high-level overview of our pipeline is presented in Figure 3.

## 3.1. Model Architecture

**Preliminaries: Arc2Face** (Paraperas Papantoniou et al., 2024) is a foundation diffusion model specializing in ID-consistent face synthesis. This is achieved through the use of ArcFace (Deng et al., 2019) as guidance via a novel conditioning approach which repurposes the original CLIP text encoder $\tau$ in Stable Diffusion (SD) to act as a facial encoder tailored to ArcFace embeddings, achieving significantly higher degree of ID similarity compared to prior methods. Given the embedding $v = \phi(x) \in \mathbb{R}^{512}$ extracted by ArcFace $\phi$ from an image $x \in \mathbb{R}^{h \times w \times c}$, $v$ replaces a placeholder token in a fixed text prompt which is then mapped by the fine-tuned encoder to a sequence in the CLIP latent space $c_{id} = \tau(v) \in \mathbb{R}^{77 \times 768}$, suitable for SD's cross-attention interface. Trained on the largest face recognition dataset (Zhu et al., 2021), Arc2Face offers strong decoupling between identity and other visual attributes, making it well-suited for our task, which aims at ID-aware, controllable portrait animation. To this end, we adopt its identity encoder and extend its pretrained 2D UNet for temporal processing and multi-source conditioning.

**Denoising Video Model.** Similar to (Guo et al., 2024; Blattmann et al., 2023b), we follow a network inflation strategy, extending Arc2Face to process latent 4D tensors $z \in \mathbb{R}^{f \times h \times w \times c}$, where $f$ denotes the temporal dimension.

After each spatial attention block in the original UNet, we insert temporal transformer blocks, consisting of self-attention layers that operate exclusively along the $f$ axis. While spatial attention processes each frame independently, treating the temporal axis $f$ as part of the batch, the subsequent temporal transformer merges the spatial dimensions $(h, w)$ into the batch axis and applies self-attention to the resulting tensor $z \in \mathbb{R}^{(h \times w) \times f \times c}$ along the $f$ axis. This enables spatially local yet temporally global information exchange among all frames, allowing the UNet to denoise a video sequence as a collection. In practice, we keep the original 2D UNet frozen and train the inserted temporal transformers, which both preserves the identity fidelity inherited from Arc2Face and limits the computational overhead.

**Audio Encoder.** To condition the model on speech input, we employ an off-the-shelf audio encoder for robust feature extraction. Following common practice (Wei et al., 2024; Xu et al., 2024a), we use wav2vec2 (Baevski et al., 2020) to extract per-frame audio embeddings, obtained by merging the final layer representations, resulting in a vector $c_a \in \mathbb{R}^{9216}$ for each frame. We then use a lightweight MLP to project them into the UNet's attention space for conditioning.

**Camera Encoder.** STARCaster supports per-frame viewpoint control for spatial synthesis. Given a camera configuration, we flatten the $4 \times 4$ extrinsics and $3 \times 3$ intrinsics matrices into a single conditioning vector $c_c \in \mathbb{R}^{25}$. Similarly, this vector is mapped to the diffusion model's attention space through a small MLP.

**Decoupled Multi-Source Cross-Attention.** We integrate disentangled control over identity, speech, and viewpoint through a decoupled multi-source cross-attention mechanism, which replaces the single-stream cross-attention used in Arc2Face, i.e., $\text{Attention}_{\text{id}}(Q, K_{id}, V_{id})$, where $Q = zW_Q$, $K = c_{id}W_{K_{id}}$, and $V = c_{id}W_{V_{id}}$, using the global identity embedding $c_{id}$ extracted by the Arc2Face encoder. Our objective is to additionally incorporate per-frame audio $c_a$ and camera $c_c$ conditions while preserving the pretrained identity attention weights. Here, we introduce parallel attention streams that share the same query but operate with trainable modality-specific key and value projections (Ye et al., 2023). The outputs from each attention path are then aggregated to form the final representation:

$$\begin{aligned} z_{\text{out}} = \ &\text{Attention}_{\text{id}}(Q, K_{id}, V_{id}) \\ &+ \lambda_a \cdot \text{Attention}_{\text{a}}(Q, K_a, V_a) \\ &+ \lambda_c \cdot \text{Attention}_{\text{c}}(Q, K_c, V_c), \end{aligned} \quad (1)$$

where $K_a = c_a W_{K_a}$, $V_a = c_a W_{V_a}$, $K_c = c_c W_{K_c}$, and $V_c = c_c W_{V_c}$. This operation is performed independently for each frame in the sequence. During training of this module, only the projection layers of the newly introduced modalities $W_{K_a}, W_{V_a}, W_{K_c}$, and $W_{V_c}$ are optimized. At inference, the

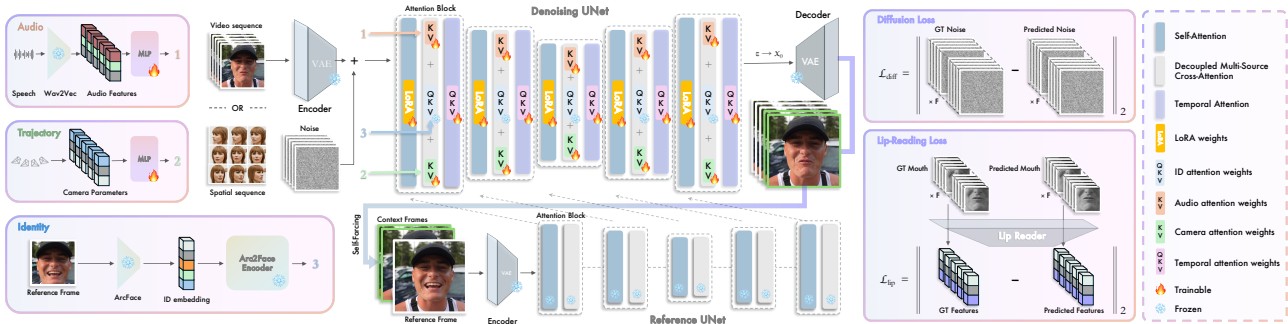

*Figure 3.* **Overview of our framework.** We extend an ID-aware backbone into a spatio-temporal autoregressive video diffusion model, which unifies ID- and audio-driven animation, reference-based synthesis, and viewpoint control. Building on the core attention block of the 2D UNet, we introduce three key extensions (Section 3.1): (1) Temporal transformer blocks for cross-frame coherence, (2) Decoupled multi-source cross-attention for integrating independent conditioning streams (identity, audio, and camera), and (3) Extended self-attention for injecting appearance features from the input or past frames via a reference encoder. During audio-visual training, we employ self-forcing-based autoregression to enhance motion diversity (Section 3.2) and a lip-reading loss to improve lip synchronization (Section 3.3). For spatial generation, we fine-tune on pseudo multi-view trajectory sequences rendered from synthetic 3D face models.

scaling factors $\lambda_a$ and $\lambda_c$ serve as hyperparameters ($[0, 1]$) to modulate the influence of each signal. Furthermore, as we will describe in Section 3.4, the audio and camera branches are trained independently in distinct stages; during each stage, the secondary modality is deactivated by setting its respective scale ($\lambda_a$ or $\lambda_c$) to zero.

**Reference Network.** Portrait animation also requires conditioning on the reference image itself. To this end, we employ a reference UNet-based framework (Cao et al., 2023), consisting of a dedicated UNet to encode the reference frame and a mutual self-attention mechanism that injects reference features into the video UNet. The core idea is that the evolving features of the generated frames should follow the same semantic and spatial structure that the reference frame would produce if processed by the same UNet layers. In practice, given the reference image, $x_{\text{ref}}$, encoded by the VAE, a frozen copy of the 2D UNet (reference network) extracts spatially aligned features $z_{\text{ref}} \in \mathbb{R}^{(h \times w) \times c}$, taken right before each self-attention module. These features are then temporally repeated and spatially concatenated with the corresponding latent representation $z$ of the evolving video frames to form:

$$z_{\text{cat}} = \text{concat}(z, z_{\text{ref}}) \in \mathbb{R}^{f \times (2h \times 2w) \times c}, \qquad (2)$$

which provides extended key and value embeddings for self-attention, $K = z_{\text{cat}}W_K$ and $V = z_{\text{cat}}W_V$, while the query features, $Q = zW_Q$, are derived directly from the video latents. This allows each frame to separately draw appearance cues from the reference image while maintaining temporal coherence through the temporal transformers. To improve adaptability between the reference and denoising UNets, we introduce lightweight learnable LoRA modules (Hu et al., 2021) into the self-attention projections $W_Q$, $W_K$, and $W_V$ of the denoising UNet. This allows the model to incorporate reference-derived features without significantly

increasing model complexity.

### 3.2. Extending Generation Length

**Autoregression.** Typically, portrait animation approaches adopt an autoregressive strategy to synthesize sequences of arbitrary length. In this setting, the model is trained to generate short video segments of fixed length $N$, conditioned on a subset of $n$ frames taken from the end of the preceding segment in the ground-truth video. We implement a similar previous token conditioning scheme through the temporal blocks of our architecture in conjunction with the reference UNet. Specifically, in addition to the global reference image, we also query the reference UNet using the most recent $n$ context frames, producing feature maps that capture both appearance and short-term temporal information. The latent representations of these context frames are prepended to the latent sequence of the current segment of $N$ frames before being processed by the video UNet. Therefore, the temporal attention layers operate on the extended sequence of length $N + n$ along the temporal axis, naturally aligning the new segment with its preceding context.

**Self-Forcing.** In typical autoregressive training (Teacher Forcing), the diffusion model denoises each video segment conditioned on clean, ground-truth context frames. This approach is known to suffer from exposure bias (Ning et al., 2023; Schmidt, 2019). During training, the model only observes perfect context, while at inference it must rely on its own predictions, leading to cumulative errors. For talking portrait generation, this is often mitigated by conditioning on the global ground-truth reference frame (the input portrait), which stabilizes appearance and pose. Yet, such strong reliance on a fixed reference often leads to uncanny animations, where the subject rarely deviates from the initial image. To overcome this, we propose adopting Self-Forcing

(Huang et al., 2025). The key idea is to perform autoregressive self-recursion during training, conditioning each segment's denoising process on the model's previous generated context frames rather than ground-truth ones. Concretely, given the predicted noise sequence $\epsilon = \{\epsilon_1, \ldots, \epsilon_N\}$ for a previous segment, we estimate the corresponding denoised frames $x_0 = \{x_{0_1}, \ldots, x_{0_N}\}$ via backward diffusion, and use the last $n$ as context for the next segment. This process can be recursively repeated for $F$ segments, with only the first segment initialized from ground-truth context. In practice, we find that even shallow recursion ($F = 2$) is sufficient to consistently improve temporal coherence, as it already doubles the effective temporal context length beyond the UNet's explicit receptive field while forcing the model to learn to correct its own generation artifacts. Using this minimal recursive depth we ensure the method remains lightweight and highly implementable on standard hardware, while still enabling more fluid, natural portrait dynamics.

### 3.3. Lip-Reading Supervision

Existing talking portrait methods (Xu et al., 2024a; Wei et al., 2024; Chen et al., 2025) are primarily trained with diffusion losses between predicted and ground-truth noise, occasionally complemented by landmark or perceptual objectives. Yet, such supervision may not sufficiently capture the fine-grained articulatory dynamics required for accurate speech synchronization. To this end, we introduce a perceptual lip-reading loss that explicitly constrains the model's speech-related motion generation. Despite similar objectives in GAN-based animation (Prajwal et al., 2020) and 3D expression reconstruction (Filntisis et al., 2022), it has not been widely adopted in diffusion frameworks, as these typically operate in a noisy latent space. We propose integrating such a loss through a supervision-by-denoising formulation, where the noisy latent is first mapped to the clean image space (see **Appendix**). Regarding the loss computation, we follow (Filntisis et al., 2022) and employ a lip-reading network (Ma et al., 2022) trained on LRS3 (Afouras et al., 2018b;a) as the perceptual encoder. For each training batch, we differentiably crop the mouth region from both the ground-truth and denoised videos using facial landmarks, convert them to grayscale, and feed them into the network to extract lip-related features from its ResNet encoder. Using a Mean Squared Error loss between the features of real and generated clips, we enforce speech alignment between ground-truth and generated videos, in addition to the photometric alignment imposed by the diffusion loss.

### 3.4. Training and Inference

**Training.** We train STARCaster in three progressive stages: **(1) Audio-Driven Motion Learning.** Only the identity and audio cross-attention streams are active ($\lambda_a = 1$, $\lambda_c = 0$), while reference conditioning is disabled. During this stage, we optimize the temporal transformer layers together with the audio-specific attention weights using photometric and lip-reading supervision on fixed-length video segments of length $N$. This stage enables the model to transition from a static identity-driven image prior to natural speech-driven facial dynamics while preserving identity consistency. **(2) Autoregressive Self-Forcing.** The reference UNet is enabled to provide reference and context conditioning, and we apply the self-forcing strategy as in Section 3.2. At this stage, we also fine-tune the added LoRA layers within the self-attention modules of the denoising UNet. This stage equips the model to generate temporally coherent sequences of arbitrary length. **(3) Temporal-to-Spatial Adaptation.** The audio stream is deactivated while we enable the camera attention branch ($\lambda_a = 0$, $\lambda_c = 1$), adapting the model for continuous view synthesis. Training remains autoregressive but now focuses on spatial consistency across trajectories. The same model weights are optimized, with camera attention layers replacing the audio-specific ones.

For stages (1) and (2), we use "in-the-wild" video datasets, including **VFHQ** (Xie et al., 2022), **CelebV-HQ** (Zhu et al., 2022), and **HDTF** (Zhang et al., 2021). We use segments of $N = 16$ frames, with $n = 2$ context frames for stage (2) and select one frame as the reference for autoregressive conditioning. The ID embedding from a random frame serves as global conditioning. For spatial training (3), since "in-the-wild" multi-view talking face data are scarce, we employ synthetic multi-view sequences by rendering smooth camera trajectories across the frontal hemisphere of 3D heads from a 3D generative model (Li et al., 2024). Training follows the same protocol, treating short trajectories as video segments with camera annotations for spatial synthesis. Owing to the large-scale video pretraining, the model rapidly adapts to the new task, enabling seamless integration of both capabilities at test time. For data processing and implementation specifics, please refer to the **Appendix**.

**Inference.** At test time, our decoupled conditions allow flexible control across multiple tasks. Using the ID embedding and a driving audio, we can synthesize novel talking videos of a person. After generating the initial segment, the final frames are used as context and reference for autoregressive continuation. To animate a specific portrait image instead, we add it as direct appearance guidance through the reference UNet, while maintaining autoregressive inference. Finally, for animations across varying views, we interpolate between the input and target viewpoints using smooth camera trajectories and animate them via the audio stream.

## 4. Experiments

We conduct comprehensive experiments to evaluate STARCaster across all supported tasks. Below, we describe the evaluation setup, metrics, and baselines for each setting.

*Table 1.* **Audio-driven portrait animation:** Quantitative comparison with state-of-the-art methods on TH-1KH (Wang et al., 2021) and Hallo3 (Cui et al., 2025). We evaluate image (FID) and video (FVD) quality, audio-visual alignment (LSE-C, LSE-D), and head motion diversity (Pose Std). **STARCaster (Ours) - ID-Driven\*** denotes ID-guided generation, i.e., without portrait conditioning. **Bold** indicates best and underline second-best result.

| | FID↓ | | FVD↓ | | LSE-C↑ | | LSE-D↓ | | Pose Std↑($\times 10^{-2}$) | |
| --- | --- | --- | --- | --- | --- | --- | --- | --- | --- | --- |
| | TH-1KH | Hallo3 | TH-1KH | Hallo3 | TH-1KH | Hallo3 | TH-1KH | Hallo3 | TH-1KH | Hallo3 |
| AniTalker (Liu et al., 2024b) | 46.75 | 32.22 | 254.68 | 166.49 | 5.419 | 5.873 | 8.818 | 8.872 | 2.339 | 2.455 |
| EchoMimic (Chen et al., 2025) | 30.71 | 18.78 | 225.19 | 151.75 | 4.551 | 5.133 | 9.682 | 9.548 | 3.534 | 3.270 |
| V-Express (Wang et al., 2024a) | 41.50 | 30.07 | 381.06 | 283.04 | 5.412 | 6.230 | 8.729 | 8.545 | 0.692 | 0.780 |
| AniPortrait (Wei et al., 2024) | 32.91 | 21.48 | 249.35 | 177.36 | 3.021 | 3.212 | 10.837 | 11.090 | 2.191 | 2.127 |
| Hallo3 (Cui et al., 2025) | 27.18 | 17.46 | 194.76 | 149.06 | 5.141 | 5.778 | 9.508 | 9.498 | 2.696 | 2.806 |
| EDTalk (Tan et al., 2024) | 44.31 | 32.87 | 277.33 | 162.63 | 5.479 | 6.282 | 8.868 | 8.585 | 0.631 | 0.682 |
| FLOAT (Ki et al., 2025) | 47.20 | 28.24 | 336.71 | 188.35 | 5.482 | 6.287 | 8.855 | 8.579 | 4.859 | 4.741 |
| **STARCaster (Ours)** | **24.89** | **16.86** | **185.24** | **145.35** | 5.493 | 6.292 | 8.724 | 8.540 | 4.896 | 4.760 |
| **STARCaster (Ours) - ID-Driven\*** | - | - | - | - | 5.627 | 6.397 | 8.695 | 8.468 | 4.905 | 4.750 |

## 4.1. Audio-Driven Portrait Animation

We first focus on audio-driven animation, where the goal is to generate a talking video from a given portrait that accurately follows the driving speech. We compare against recent open-source talking head models (Liu et al., 2024b; Chen et al., 2025; Wang et al., 2024a; Wei et al., 2024; Cui et al., 2025; Tan et al., 2024; Ki et al., 2025) using two high-quality datasets, specifically **TalkingHead-1KH (TH-1KH)** (Wang et al., 2021) and **Hallo3** (Cui et al., 2025). From each dataset, we sample 100 clips and use all methods to reconstruct them by animating the first frame using the corresponding audio. All videos are converted to 25 FPS for evaluation. We assess performance along three main axes - visual quality, lip synchronization, and motion expressiveness - using the following metrics: (1) Fréchet Inception Distance **(FID)** (Heusel et al., 2017) for image quality, (2) 16 frames Fréchet Video Distance **(FVD)** (Unterthiner et al., 2018) for temporal consistency, (3) Lip-Sync Error **(LSE-D)** and Confidence **(LSE-C)** (Prajwal et al., 2020) for audio-visual alignment, (4) **Pose Std**, the standard deviation of FLAME pose parameters (Li et al., 2017) extracted with (Feng et al., 2021), as a measure of head motion diversity.

As shown in Table 1, STARCaster achieves the lowest FID and FVD scores, even surpassing Hallo3 (Cui et al., 2025), a substantially larger 5B-parameter DiT model, despite being evaluated on clips from Hallo3's own training data. Our model further exhibits superior lip synchronization and greater pose diversity, driven by the proposed recursive training and lip-reading supervision. Visual comparisons in Figure 4 further demonstrate that baselines often struggle to capture accurate lip motion, whereas STARCaster produces mouth shapes closer to the ground-truth, indicating better alignment to the driving audio. In addition, we compare identity similarity against the top three performing methods in terms of head motion in Figure 5, where we achieve stronger consistency with the reference subject, benefiting from the Arc2Face prior and our decoupled conditioning design that prevents interference between speech and identity. Finally, for perceptual validation, we conducted a user study

with 35 participants who were asked to select the animation that appeared most natural in terms of head dynamics across 25 randomly selected sets of samples. As reported in Figure 6, STARCaster was preferred in the majority of cases, confirming its superior realism.

Under the same setup, we also assess STARCaster's stochastic ID-driven video generation, where the ID embedding is used as conditioning instead of the portrait image. For fairness, we focus on lip sync and pose diversity, as FID and FVD inherently favor image-driven methods. As reported in Table 1 (last row), our ID-driven generations achieve competitive motion diversity and even better lip sync than the image-driven case, demonstrating excellent performance even when animating subjects in novel contexts.

Finally, to illustrate the inference efficiency of our UNet-based architecture, Table 2 reports the sampling time required to generate a 5-second talking video clip at 25 FPS, in comparison with representative diffusion-based baselines. Our method achieves the lowest inference time, thanks to its lightweight design enabled by careful adjustments to the image backbone. In contrast, the 5B-parameter DiT-based approach used in Hallo3 is approximately an order of magnitude slower, highlighting the computational overhead of recent large-scale DiT models for this task.

*Table 2.* Inference time (using an NVIDIA A100 GPU) of diffusion-based methods for generating a 5-second video clip at 25 FPS.

| **Ours** | V-Express (Wang et al., 2024a) | EchoMimic (Chen et al., 2025) | AniPortrait (Wei et al., 2024) | Hallo3 (Cui et al., 2025) |
| --- | --- | --- | --- | --- |
| **2.2 min** | 2.5 min | 2.7 min | 3.7 min | 21.4 min |

## 4.2. 3D-Aware Talking Portrait Generation

We then evaluate animation across novel views using the NeRSemble dataset (Kirschstein et al., 2023), which contains multi-view talking videos of actors captured from 16 calibrated viewpoints. We sample 100 identities, using one random camera view as the reference image and three other views as target videos. We compare against recent 3D-aware

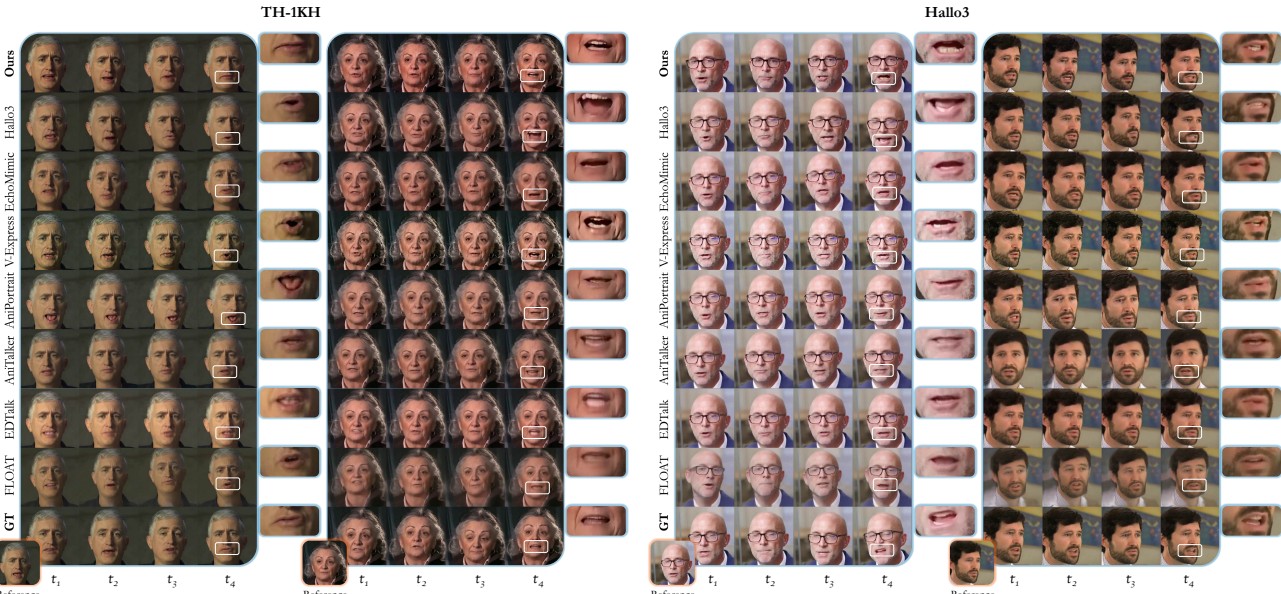

*Figure 4.* Visual comparison with recent talking portrait methods (Liu et al., 2024b; Chen et al., 2025; Wang et al., 2024a; Wei et al., 2024; Cui et al., 2025; Tan et al., 2024; Ki et al., 2025) on the TH-1KH (Wang et al., 2021) and Hallo3 (Cui et al., 2025) datasets.

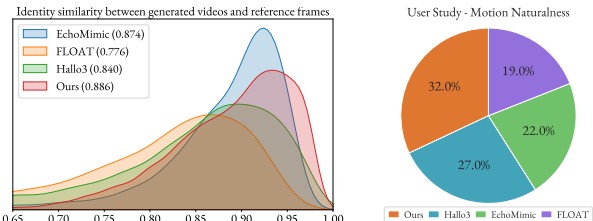

*Figure 5.* Distribution (and mean) of cosine similarity scores between reference ID and generated frames.

*Figure 6.* Users' preference on the naturalness of head motion in generated videos.

*Table 3.* **3D-aware talking portrait generation:** Quantitative comparison with 3D-aware talking portrait methods on the NeRSemble dataset based on video quality (FID, FVD), reconstruction accuracy across novel views (PSNR, SSIM, LPIPS) and audio-visual alignment (LSE-C, LSE-D). **Bold** indicates best and underline second-best result.

| | FID↓ | FVD↓ | LPIPS↓ | SSIM↑ | PSNR↑ | LSE-C↑ | LSE-D↓ |
|---|---|---|---|---|---|---|---|
| GOHA (Li et al., 2023b) | 85.63 | 339.52 | 0.614 | 0.354 | 8.487 | 3.070 | 8.322 |
| Portrait4D-v2 (Deng et al., 2024) | 48.91 | 243.91 | 0.560 | 0.569 | 12.469 | 3.384 | 8.404 |
| Real3DPortrait (Ye et al., 2024) | 32.33 | 240.11 | 0.608 | 0.545 | 10.289 | 3.401 | 7.967 |
| **STARCaster (Ours)** | **29.52** | **222.89** | **0.477** | **0.577** | **14.219** | **3.957** | **7.952** |

talking portrait approaches. Since audio-driven baselines are limited (Ye et al., 2024), we also include expression-driven methods (Li et al., 2023b; Deng et al., 2024). For each, we generate 300 view-conditioned animations and evaluate against the ground-truth videos using FID and FVD for visual and temporal quality, as well as LSE-C, LSE-D for audio-visual alignment. For view consistency, we use spatially aligned metrics (PSNR, SSIM, LPIPS) between corresponding target views. As shown in Table 3, STARCaster achieves clear improvements in image quality, lip synchronization and, most importantly, view accuracy, despite being conditioned on audio instead of driving videos, underscoring the effectiveness of our unified spatio-temporal design that combines video-based spatial priors with lightweight synthetic multi-view supervision. For qualitative results of novel-view animations, see the **Appendix**.

### 4.3. Attention Visualization

Our model employs a decoupled attention mechanism to integrate multiple conditioning signals. In Figure 7, we visualize the attention maps from a representative cross-attention layer of the UNet during spatio-temporal video generation, providing insight into how different modalities influence the generation process. Specifically, audio attention is highly localized, focusing on speech-relevant regions such as the lips and eyelids. In contrast, camera attention is more spatially distributed, reflecting the global impact of viewpoint changes on both foreground and background. Notably, stronger responses are observed near facial contours, which are more sensitive to pose variations than interior facial regions. Finally, identity attention seems to affect both the face and surrounding areas. While the response in the background may appear counterintuitive, it originates from the Arc2Face backbone, which relies solely on identity features to guide full-image synthesis. Furthermore, ID embeddings may inadvertently encode certain non-identity attributes, such as accessories or subtle contextual cues around the face, which is a known limitation of Face Recognition features. Overall, while attention is computed in latent space and the visualized maps (upsampled to the orig-

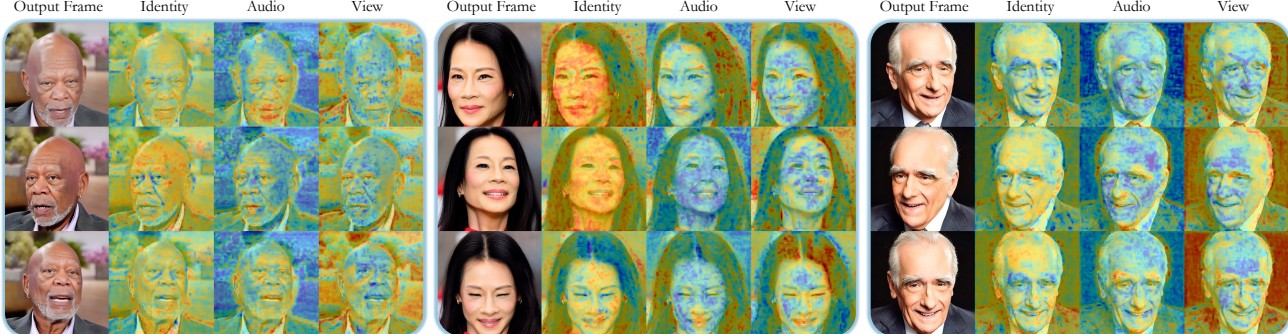

*Figure 7.* **Decoupled Multi-Source Cross-Attention:** Generated animations and corresponding per-frame spatial attention maps for each conditioning signal (identity, audio, camera). For each spatial location, we visualize the maximum cross-attention weight over the tokens of the corresponding modality as a heatmap, highlighting regions that attend most strongly to that source.

inal image space for clarity) provide only an approximate interpretation, they still offer useful evidence that each attention branch captures the intended information, particularly highlighting the localized behavior of audio features.

### 4.4. Ablation Studies

We further assess the contribution of key design components using the same evaluation setup as in Section 4.1. Specifically, we compare the full STARCaster against three variants: (1) without the lip-reading loss, (2) without recursive training, i.e., training with teacher forcing, where each segment is denoised using clean context frames, and (3) without progressive audio-visual training, i.e., replacing the first two stages with a single autoregressive stage. As shown in Table 4, all components lead to improved performance, confirming their importance for achieving natural, synchronized motion. A visual comparison in Figure 8 further illustrates the impact of the lip-reading loss.

*Table 4.* Ablation studies on TH-1KH and Hallo3 datasets. **Bold** denotes best and underline second-best performance.

| | **FVD**↓ | | **LSE-C**↑ | | **LSE-D**↓ | | **Pose Std**↑($\times 10^{-2}$) | |
| --- | --- | --- | --- | --- | --- | --- | --- | --- |
| | TH-1KH | Hallo3 | TH-1KH | Hallo3 | TH-1KH | Hallo3 | TH-1KH | Hallo3 |
| **STARCaster** | **185.24** | **145.35** | **5.493** | **6.292** | **8.724** | **8.540** | **4.896** | **4.760** |
| w/o lip-reading | 188.46 | 147.11 | 5.140 | 5.929 | 9.016 | 8.794 | 4.630 | 4.567 |
| w/o self-forcing | 197.28 | 148.06 | 5.294 | 6.112 | 8.936 | 8.659 | 4.386 | 4.161 |
| w/o progr. training | 191.46 | 152.98 | 5.342 | 6.233 | 8.789 | 8.611 | 4.695 | 4.361 |

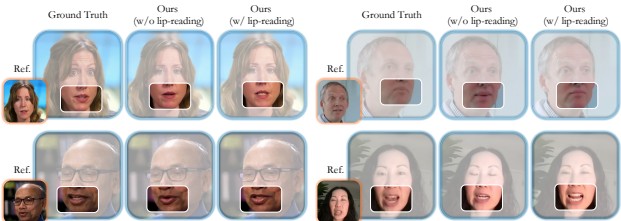

*Figure 8.* Audio-driven animations of reference portraits. The same target frame is shown for both variants. The model trained with the lip-reading loss produces more accurate lip movements.

## 5. Conclusion

We introduced STARCaster, a diffusion-based framework for ID- and view-aware talking portraits, without relying on explicit 3D representations. Our spatio-temporal model combines monocular "in-the-wild" videos and synthetic multi-view sequences to learn coherent motion and view-consistent generation. Lip-reading supervision enhances audio-visual synchronization, while a self-forcing-based autoregressive training strategy enhances vividness. Moreover, formulating novel-view portrait synthesis as autoregressive sequence generation conditioned on the reference frame yields more accurate view transitions than in 3D inversion-based methods. Comprehensive experiments demonstrate that STARCaster achieves state-of-the-art speech-driven animation across different settings, advancing diffusion-based facial video generation toward unified 3D-aware modeling.

**Limitations.** While our approach advances portrait animation, some limitations remain. First, our model - like most diffusion-based frameworks - suffers from inherently slow inference speeds. It currently falls short of the real-time performance achieved by traditional GAN-based alternatives. Regarding architectural and data scope, our method is specifically designed for portrait-level animation; as such, background dynamics are beyond the scope of this work. Furthermore, the supported range of camera control is restricted to frontal and near-profile views. This constraint aligns with the typical requirements of talking-head scenarios and reflects the distribution of available audio-visual training datasets. Consequently, $360°$ coverage and rear-view synthesis are not supported. Crucially, our objective is not to provide unrestricted 3D rendering, but rather to demonstrate that view synthesis can be effectively reformulated as a conditional video generation task. Finally, fine-tuning on synthetic 3D data introduces a slight smoothing effect on skin textures and occasional color inconsistencies during rapid camera transitions due to the domain gap. These artifacts primarily stem from imperfections in the synthetic data rather than the architectural design itself.

**Acknowledgements.** S. Zafeiriou and part of the research was funded by the EPSRC Project GNOMON (EP/X011364/1) and Turing AI Fellowship (EP/Z534699/1). RA Potamias was partially supported from Project GNOMON (EP/X011364/1). B. Kainz received support from the ERC, project MIA-NORMAL 101083647, DFG 512819079, and by the State of Bavaria (HTA). HPC resources were provided by the Erlangen National High Performance Computing Center (NHR@FAU) of the Friedrich-Alexander-Universität Erlangen–Nürnberg (FAU) under the NHR project b180dc. NHR@FAU hardware is partially funded by the German Research Foundation (DFG) - 440719683.

## Impact Statement

We acknowledge potential ethical concerns, as controllable facial animation technologies may be misused to create deceptive or non-consensual content. While our work is intended to support positive applications in accessibility and creative media, we recognize these risks and emphasize the importance of complementary safeguards, such as reliable synthetic content detection and responsible deployment practices.

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

# A. Appendix

This appendix provides additional details of the proposed approach, including (Appendix A.1):

- data collection and preprocessing

- architectural components

- training and inference hyperparameters

- design of lip-reading loss and recursive training

as well as additional qualitative results (Appendix A.2).

## A.1. Implementation Details

### A.1.1. TALKING VIDEO DATA

As described in Section 3.4, the first two training stages rely on large-scale "in-the-wild" talking video datasets. We use a mixture of publicly available data sources (Xie et al., 2022; Zhu et al., 2022; Zhang et al., 2021), consisting primarily of English-speaking videos, totaling roughly 11M frames. We apply a multi-step preprocessing and cleaning pipeline. First, we perform face detection to remove frames without a detected face. To ensure the data corresponds to single-person talking portraits, we further discard frames containing multiple faces (approximately 0.4M frames). Since our model does not explicitly model hand motion, frames where hands occlude the subject often introduce artifacts. To address this, we use Grounding DINO (Liu et al., 2024a) to discard such frames (around 1.8M frames). The effect of this filtering process is illustrated in Figure 9. All videos are then cropped and resized to $512 \times 512$, aligning faces to an FFHQ-style (Karras et al., 2019) template. Then, we extract audio features for each video using wav2vec2 (Schneider et al., 2019; Baevski et al., 2020) and ArcFace identity embeddings via insightface. During training, we randomly sample subsequences of length $N = 16$ from the cleaned clips.

### A.1.2. SYNTHETIC MULTI-VIEW DATA

For creating spatial training data (Stage 3), we employ a state-of-the-art 3D-aware face generator (Li et al., 2024). We generate 20K 3D heads and render random smooth oscillatory camera trajectories spanning $140°$ in azimuth and $70°$ in elevation across the frontal hemisphere, matching typical pose ranges in real datasets like FFHQ. We follow the standard convention in (Li et al., 2024), where heads are centered at the origin, and cameras lie on a sphere (radius 2.7m, FOV 18.837°), facing inward. This rendering setup also ensures alignment with the FFHQ crop used for real video training. Each trajectory is rendered for 5 seconds at 30 FPS, producing approximately 3M synthetic frames

in total. Examples of such trajectories are shown in Figure 10. As in the audio-visual training stages, we sample segments of length $N = 16$ during optimization. These view sequences are used as standard training clips, with the corresponding cam2world extrinsics as per-frame conditions. During inference, for viewpoint control, we first align the input image and estimate its initial camera pose via an off-the-shelf estimator as in (Li et al., 2024). Then, target azimuth/elevation trajectories are defined as sequences of absolute camera matrices starting from the initial pose.

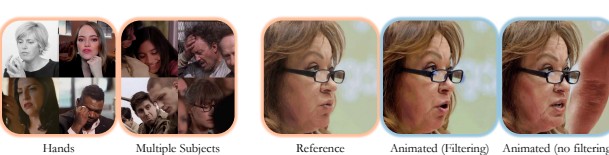

*Figure 9.* **Effect of video data cleaning.** (**Left**) Example frames discarded from the talking video datasets due to hands or multiple subjects. (**Right**) A model trained on the unfiltered data exhibits significant artifacts in audio-driven animations of reference portraits.

### A.1.3. MODEL ARCHITECTURE

Our model builds on Arc2Face (Paraperas Papantoniou et al., 2024), which employs a fine-tuned UNet and an identity encoder derived from stable-diffusion-v1-5. Our temporal transformer blocks introduce an additional 0.5B parameters on top of the original UNet's 0.9B parameters. For audio and camera conditioning, we add separate key/value projection matrices to the UNet's cross-attention layers, along with two lightweight two-layer MLPs for feature projection, resulting in 0.2B additional parameters. For autoregressive training, we maintain a copy of Arc2Face's UNet as the reference network, while we augment the main UNet with small LoRA modules applied to the self-attention layers, using a rank of 64. Importantly, in contrast to all prior works that train the reference network (initialized from a generic T2I backbone), we keep it frozen, as we found its identity-tailored weights (initialized from Arc2Face) sufficient to capture the reference appearance without further tuning. Our reference network, thus, introduces no additional memory (shared weights with the spatial layers of the denoising UNet), whereas competing works inherently use different checkpoints for the two networks. Overall, our model remains significantly more efficient than recent DiT baselines (10× faster).

### A.1.4. TRAINING AND INFERENCE

All models were trained on 8 NVIDIA H200 GPUs using AdamW (Loshchilov & Hutter, 2017) with a learning rate of 1e-4, a per-GPU batch size of 2, and 4 gradient accumulation steps per GPU. For the first stage (**Audio-Driven Motion Learning**), we pretrain on VFHQ (Xie et al., 2022)

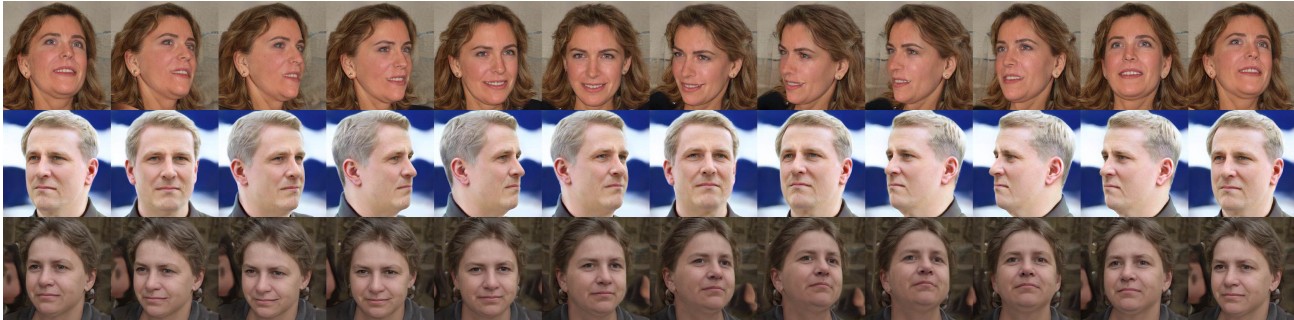

*Figure 10.* Examples of synthetic trajectories from our dataset used for spatial fine-tuning. Each row shows a subsampled version of a smooth oscillatory trajectory of 5 seconds at 30 FPS.

and CelebV-HQ (Zhu et al., 2022) for 200K iterations to cover diverse "in-the-wild" scenarios, followed by a 10K-iteration fine-tuning phase on HDTF (Zhang et al., 2021) to focus on high-quality talking portraits. The lip-reading loss is activated after 100K iterations, once the model begins producing reasonable lip motion. Diffusion and lip-reading losses are weighted equally (1.0 each). In the second stage (**Autoregressive Self-Forcing**), we fine-tune on HDTF for 6K iterations using the self-forcing-based scheme. In the third stage (**Temporal-to-Spatial Adaptation**), we further fine-tune the model on our synthetic multi-view dataset for an additional 10K steps. At each stage, we also apply modality dropout with a probability $p = 0.05$ as part of classifier-free guidance, by randomly masking the corresponding conditioning signal (audio or camera). For inference, we adopt DPM-Solver (Lu et al., 2022a;b) with 25 denoising steps and a classifier-free guidance scale of 3.0. Our model can generate videos of up to 30 FPS.

### A.1.5. LIP-READING LOSS

The pretrained lip-reading network (Ma et al., 2022) operates in pixel space, whereas our diffusion backbone inherits the VAE from Stable Diffusion (SD) and, thus, works in a compressed latent space. Consequently, during training we first decode the predicted video latents back into pixel space. We then perform on-the-fly landmark detection to localize and crop the mouth region before passing it to the lip-reading network. While these operations - including landmark detection, alignment, and feature extraction - are performed on the GPU, decoding the VAE latents into pixel space inevitably slows down training and increases memory consumption during backpropagation. To mitigate this overhead, we employ two strategies. First, since the lip-reading network only requires the mouth region, we avoid decoding full frames. Specifically, we retain a square 60% spatial crop of each latent frame, leveraging the FFHQ-style alignment of our training data, which constrains the mouth location. Second, we apply *VAE slicing*, decoding the cropped latents one frame at a time to further reduce

memory usage. These optimizations make training with the lip-reading loss feasible, albeit with a reduced batch size, as reported above. Notably, the selected lip-reading network is relatively lightweight (40M parameters) compared to the diffusion model, and the loss is applied only during a subset of the **Audio-Driven Motion Learning** stage, further limiting its impact on the overall training cost.

### A.1.6. RECURSIVE TRAINING

Similarly, during the **Autoregressive Self-Forcing** stage, the recursive formulation introduces additional memory overhead. Specifically, the final frames of a denoised segment are reused as conditioning inputs for denoising the subsequent segment, which complicates backpropagation through time. As a result, under our computational constraints, we limit the recursion depth to $F = 2$ segments, which already provides consistent gains (Table 4). Deeper recursion for even further improvement can be achieved at additional cost, i.e., by further reducing the batch size, or lowering the training resolution (e.g., $480 \times 480$ instead of $512$). Finally, it is important to note that our diffusion model adopts an $\epsilon$-prediction strategy, i.e., it predicts the noise $\epsilon$ added to the input. However, both the lip-reading loss and recursive training require a denoised estimate $x_0$ of the current noisy video segment. For efficiency, we therefore adopt a one-step prediction strategy, applying the standard backward diffusion step to remove the estimated noise $\epsilon$ and obtain the clean prediction $x_0$. Although this yields a less accurate (e.g., blurrier) reconstruction compared to multi-step sampling methods such as DDIM, we find this faster one-step approach sufficient for extracting lip features and providing context frames for recursive continuation. Note that both the lip-reading supervision and the self-forcing strategy are adopted at later training stages, at which point the model already predicts reasonable videos in one step even at the highest noise levels, e.g., $t = 1000$ (see Figure 11).

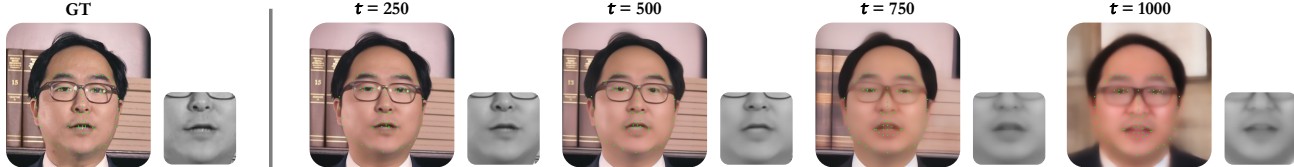

*Figure 11.* A random video frame and its single-step reconstruction from our diffusion model at increasing timesteps (noise levels), after the first 100K training iterations, at which point the lip-reading loss is activated. For each timestep, alongside the full frame, we additionally visualize the predicted 2D landmarks and the corresponding grayscale mouth crops used as input to the lip-reading network. Even at the highest noise level ($t = 1000$), where reconstructions become noticeably blurry, the model preserves sufficient structural information around the mouth region to produce meaningful lip features for supervision. Please zoom in for details.

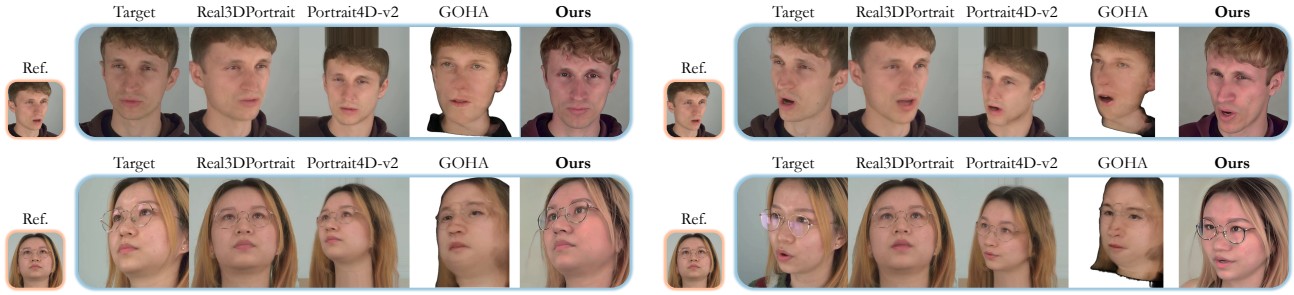

*Figure 12.* Visual comparison with 3D-aware methods (Ye et al., 2024; Li et al., 2023b; Deng et al., 2024) on the NeRSemble dataset (Kirschstein et al., 2023), showing animations from different views.

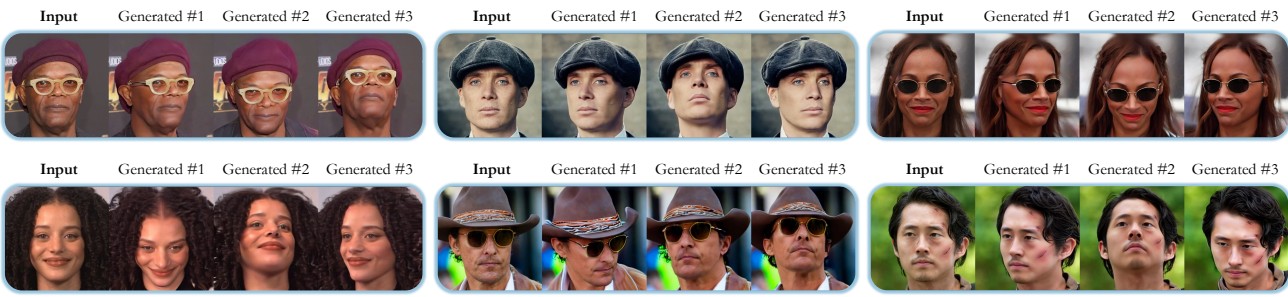

*Figure 13.* Examples of camera viewpoint control applied to input portraits using our spatio-temporal model.

## A.2. Additional Qualitative Results

### A.2.1. 3D-AWARE PORTRAIT ANIMATION

As discussed in Section 4.2, our method outperforms 3D-aware talking portrait baselines on the NeRSemble evaluation. The qualitative comparison provided in Figure 12 further illustrates that prior methods either fail to change viewpoint (Ye et al., 2024) or introduce strong artifacts (Li et al., 2023b; Deng et al., 2024), while our approach delivers more stable and accurate audio-visual animations across views. Further "in-the-wild" examples of novel view synthesis in Figure 13 demonstrate that our method maintains stable camera control under challenging real-world scenarios (e.g., hats, glasses), thanks to the robust appearance prior learned during video pretraining.

### A.2.2. ID-DRIVEN TALKING PORTRAIT GENERATION

Finally, in Figures 14 and 15, we provide examples of diverse speech-driven animations produced by STARCaster, using only identity features. Our model generalizes effectively across subjects, enabling flexible and recontextualized animations.

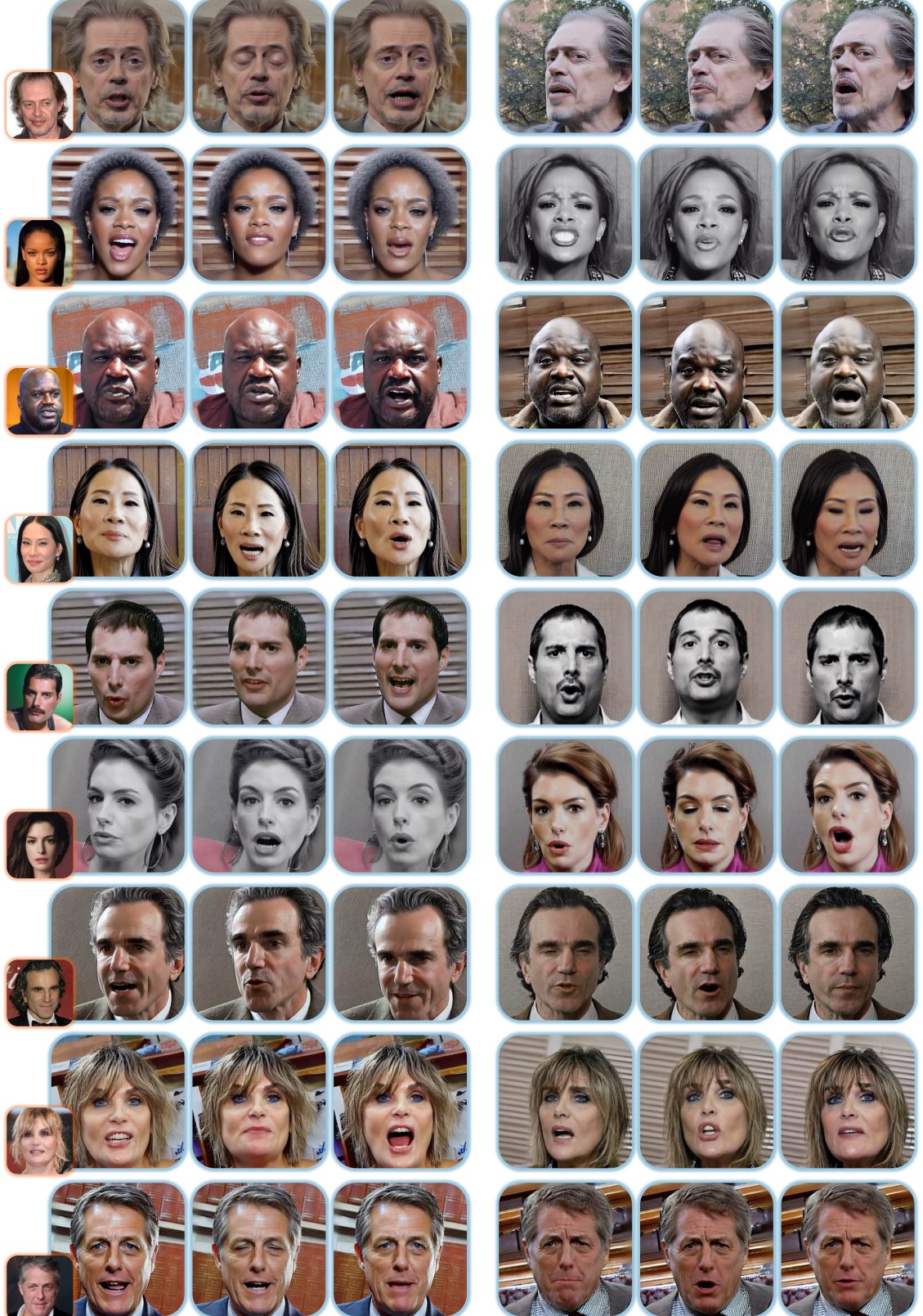

*Figure 14.* Diverse animations generated by our model, conditioned on the input identity and a driving audio.

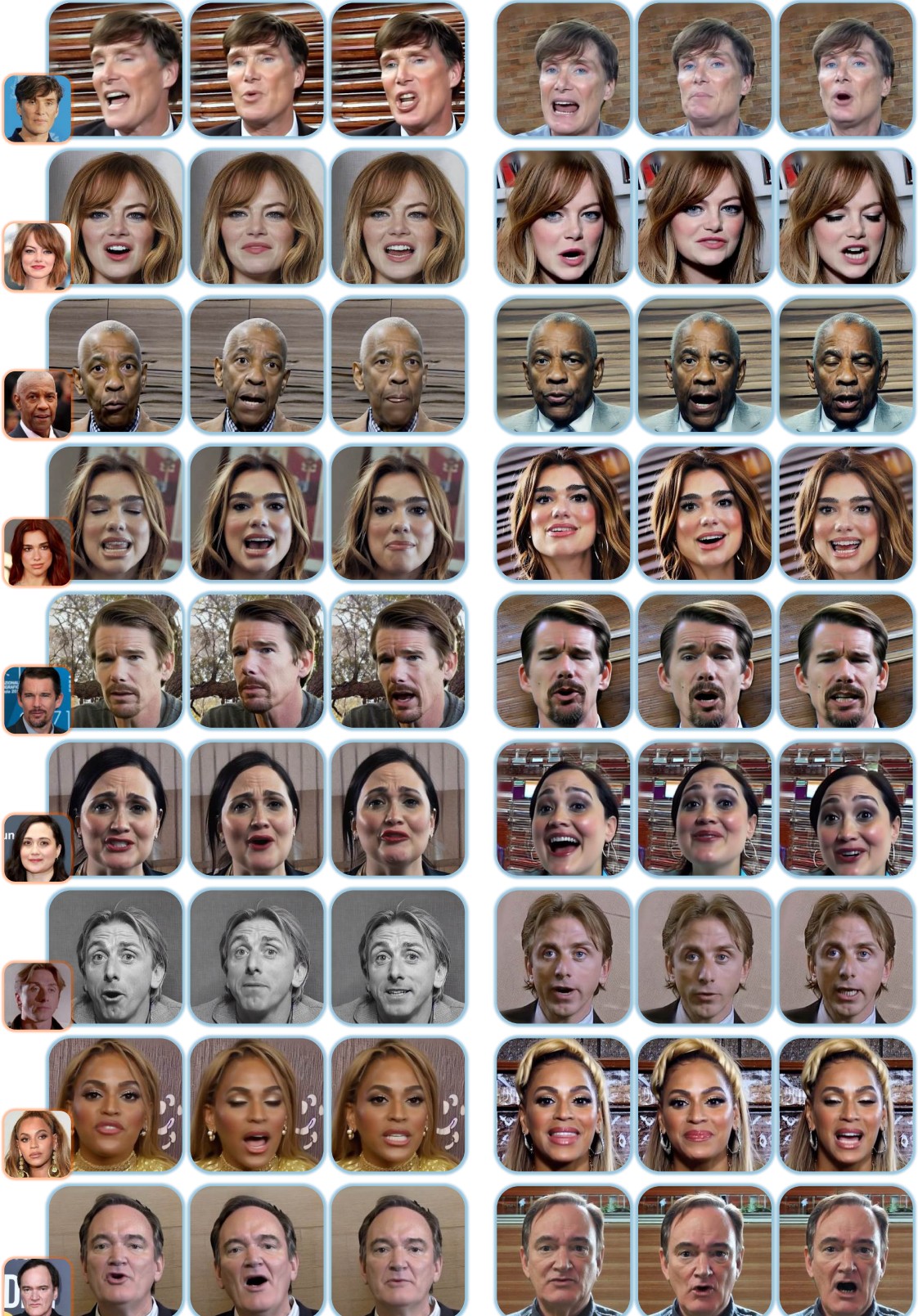

*Figure 15.* Diverse animations generated by our model, conditioned on the input identity and a driving audio (cont.).

