# OpenReview forum: "STARCaster: Spatio-Temporal AutoRegressive Video Diffusion for Identity- and View-Aware Talking Portraits"
_ICML.cc/2026/Conference — ICML 2026 regular_

### Official Review · Reviewer_HYn1 · 2026-02-20

**Soundness:** 3
**Presentation:** 4
**Significance:** 4
**Originality:** 3
**Overall Recommendation:** 5
**Confidence:** 5

**Summary:**

This paper presents STARCaster, a unified video diffusion framework for identity-aware and free-viewpoint talking portraits. It addresses the limited motion diversity of 2D methods and the identity drift common in 3D-aware animation. The authors introduce softer identity constraints to enhance motion variety and leverage synthetic multi-view data to achieve implicit 3D awareness without explicit 3D representations. Experimental results and high-quality demos effectively validate the model's superiority in visual fidelity, lip-sync, and cross-view coherence.

**Compliance With Llm Reviewing Policy:**

Affirmed.

**Key Questions For Authors:**

- Explain why a UNet-based model is chosen instead of a DiT-based model, and clarify why an image-to-video pretrained model is not directly adopted as the base model.

- Explain the reason for the color changes.

- Discuss and compare GAN-based methods (e.g., EDTalk).

**Limitations:**

yes

**Strengths And Weaknesses:**

Strengths:
- The paper provides a clear and insightful analysis of the limitations in existing 2D and 3D talking head paradigms, offering well-motivated technical solutions to address motion diversity and identity drift.
- The manuscript is well-written, and the accompanying video demonstrations are high-quality, effectively validating the model's capabilities in various scenarios.
- Unlike many existing methods that fine-tune pre-trained image-to-video models, this work builds upon a specialized image generation prior (Arc2Face) and introduces "softer" identity constraints during video training to enhance motion variety
- The framework successfully unifies camera control, identity-aware generation, and audio-driven animation within a single, coherent pipeline

weakness:
- The model is built on a U-Net backbone. Given that Diffusion Transformers (DiT) are increasingly demonstrating superior scaling and performance in video generation, the authors should discuss the trade-offs or consider using a DiT-based architecture to remain competitive with the current state-of-the-art
- Based on the provided qualitative results, there appear to be noticeable color shifts or inconsistencies in the generated frames compared to the original reference images
- The use of cross-attention to integrate identity, audio, and camera signals is a relatively standard technique in the field. The technical contribution in this specific area may be seen as incrementa
- Since the paper explicitly claims that GAN-based models lack diversity, the evaluation should include a broader range of recent GAN-based methods—such as EDTalk—to provide a more comprehensive discussion and experimental comparison regarding motion variety

---

> ### Author Rebuttal · Authors · 2026-03-30
>
> We thank the reviewer for the strong support and insightful feedback. We are pleased that the reviewer finds our video demonstrations high-quality and appreciates our specialized identity-aware approach beyond the standard image-to-video paradigm. Below, we present our responses to the reviewer’s comments.
>
> **1. Comparison with GAN baseline (EDTalk)**
>
> >We thank the reviewer and include the comparison:
>
> >| Method | FID↓ (TH) | FID↓ (Hallo3) | FVD↓ (TH) | FVD↓ (Hallo3) | LSE-C↑ (TH) | LSE-C↑ (Hallo3) | LSE-D↓ (TH) | LSE-D↓ (Hallo3) | Pose Std↑ (TH) | Pose Std↑ (Hallo3) |
> >|--------|:---------------:|:----------------:|:----------------:|:----------------:|:------------------:|:------------------:|:------------------:|:------------------:|:----------------------------:|:----------------------------:|
> >| EDTalk | 44.31 | 32.87 | 277.33 | 162.63 | 5.479 | 6.282 | 8.868 | 8.585 |  0.631 | 0.682 |
> >| **Ours** | **24.89** | **16.86** | **185.24** | **145.35** | **5.493** | **6.292** | **8.724** | **8.540** | **4.896** | **4.760** |
>
> >While GAN baselines offer strong efficiency (training and inference) and competitive lip-sync, they are fundamentally limited by their latent representation capacity. This results in **noticeably lower visual fidelity (higher FID/FVD)**, as well as artifacts such as warped facial dynamics and foreground–background inconsistencies. Moreover, their motion diversity is significantly constrained, as reflected by the **substantially lower pose variation**.
>
> **2. Technical contribution beyond cross-attention**
>
> >Our key technical contribution lies in proposing **a unified model to address different tasks (audio-driven animation, identity recontextualization, and controllable view synthesis)**. Prior works address these tasks **independently**, often requiring incompatible pipelines (2D diffusion vs. 3D inversion).
>
> >Importantly, our model enables **stochastic, identity-driven generation**, unlike prior approaches that require reference images. Moreover, aligned with `WZ6a`'s and `kg1S`’s observations, we propose a **new paradigm for spatial view synthesis** that “bypasses complex and artifact-prone 3D inversion”, while achieving strong empirical performance in real data.
>
> >Overall, while we agree that individual components, such as cross-attention, are known, we emphasize that:
>
> >- the **unified formulation**
> >- the **non-trivial adaptation of components**
> >- and the **consistent empirical gains across tasks**
>
> >constitute a meaningful contribution.
>
> **3. Color inconsistency**
>
> >Some color inconsistencies during camera control mainly stem from imperfections in the synthetic training data (see also our response to `WZ6a` regarding the domain gap). Additionally, our model inherits the typical saturation-speed trade-off of diffusion models: aggressive guidance scales or fewer inference steps can introduce color shifts. These effects can be mitigated by adjusting inference settings. We will include this discussion in the revision for clarity and transparency.
>
> **4. UNet vs DiT**
>
> >Our choice is motivated by:
>
> >- **efficiency** (10× faster than DiT baselines)
> >- compatibility with **Arc2Face identity prior**
>
> >While DiT scales well for general video, our results show that a **carefully adapted UNet can outperform larger DiT models** in audio-driven animation, while **allowing ID-to-Video generation** beyond just I2V supported by the baselines.

---

> > ### Author Rebuttal · Reviewer_HYn1 · 2026-04-01
> >
> > My concerns (Unet/color/EDTalk) have been adequately addressed.

---

> > > ### Author Response · Authors · 2026-04-06
> > >
> > > Thank you for your insightful feedback and for the constructive review process. We sincerely appreciate your time and valuable comments. We are pleased that our rebuttal adequately addressed your concerns, and we are grateful for your recognition of our work.

---

### Official Review · Reviewer_kg1S · 2026-03-11

**Soundness:** 2
**Presentation:** 2
**Significance:** 2
**Originality:** 2
**Overall Recommendation:** 4
**Confidence:** 4

**Summary:**

The paper presents STARCaster, a unified spatio-temporal autoregressive video diffusion model designed for both audio-driven talking portrait animation and free-viewpoint synthesis. Moving away from explicit 3D representations, the method builds upon an identity-aware image backbone (Arc2Face), extending it with temporal transformers and a Decoupled Multi-Source Cross-Attention mechanism to independently process identity, audio, and camera conditioning. The training pipeline proceeds in three stages: audio-driven motion learning, autoregressive self-forcing to mitigate exposure bias, and temporal-to-spatial adaptation using synthetic multi-view data.

**Compliance With Llm Reviewing Policy:**

Affirmed.

**Final Justification:**

Thank the authors for their response. Most of my concerns have been resolved. Two minor issues remain: the user study lacks statistical significance tests, and the self‑forcing depth (F=2) does not fully validate long‑term exposure bias over very long sequences. Nevertheless, considering the rebuttal and other reviewers’ feedback, I will raise my score.

**Key Questions For Authors:**

1. The lip-reading loss relies on a one-step backward diffusion approximation of  At high noise levels (t near T),  is almost purely guessed, resulting in extreme blurriness. How do you prevent the lip-reading network from backpropagating meaningless gradients during these early diffusion steps? Have you considered applying the lip-reading loss only at lower noise levels?

2. During Temporal-to-Spatial Adaptation (Stage 3), the audio stream is turned off. Could you provide the LSE-C and LSE-D scores for the NeRSemble 3D evaluation?

3. You acknowledge that computational limits cap the self-forcing depth to F=2. Given this shallow depth, how much of the FVD improvement (Table 3) is actually due to long-term consistency vs. simple short-term smoothing? Can you provide FVD metrics evaluated specifically on very long sequences to prove the exposure bias is truly mitigated?

4. Equation (1) shows a direct summation of three independent cross-attention outputs. Did you experiment with adaptive gating to fuse these signals?

**Limitations:**

The authors must explicitly state the memory bottleneck constraint in the main paper (currently hidden in Appendix A.3), acknowledging that the hardware demands of unrolling diffusion models severely limit the self-forcing recursion depth to F=2. Furthermore, they must explicitly state the geometric boundaries of their view synthesis (limited to the  frontal hemisphere based on the synthetic data generation ) rather than claiming unbounded "free-viewpoint" generation.

**Strengths And Weaknesses:**

Strengths:

1. Spatial view synthesis is simply viewed as another form of temporal video generation (spatiotemporal adaptation), cleverly bypassing the complex optimizations and artifact-prone inversion steps required for explicit 3D perception models.

2. This paper has a clear structure and logic, and the effectiveness of the proposed method is verified on multiple datasets.

Weaknesses:

1. A major contribution claimed is "long-term temporal consistency" via Autoregressive Self-Forcing. However, Appendix A.3 reveals that due to memory constraints, the recursion depth is limited to merely F=2. Calling a recursion depth of 2 "long-term" modeling is a significant overstatement. If the architecture is too heavy to unroll beyond two segments, the practical value of the self-forcing mechanism is fundamentally compromised.

2. In Stage 3 (Temporal-to-Spatial Adaptation), the authors explicitly state that "the audio stream is deactivated". Neural Networks are highly prone to catastrophic forgetting when a previously learned condition is deactivated during fine-tuning. Crucially, in the 3D-aware evaluation (Table 2), the authors do not report any lip-sync metrics (LSE-C or LSE-D) for the novel view synthesis on the NeRSemble dataset. There is zero quantitative proof that the model maintained its lip-sync capabilities after Stage 3.

3. The paper claims "free-viewpoint talking portraits". However, the spatial adaptation relies entirely on synthetic trajectories spanning only  in azimuth and  in elevation. Calling it "free-viewpoint" is misleading when the operational envelope is strictly limited to the frontal hemisphere.

4. The manual assessment relied on only 20 participants evaluating 15 samples. This yielded only 300 data points, which is statistically inadequate.

---

> ### Author Rebuttal · Authors · 2026-03-30
>
> We sincerely thank the reviewer for the detailed and constructive feedback.
>
> **1. Self-forcing depth (F=2)**
> >We agree that “long-term” may overstate the current setting and will revise the wording accordingly. That said, even shallow self-forcing (F=2) already provides consistent gains (Table 3). Deeper recursion is feasible at additional cost: F=3 can be achieved with smaller batch size, and F=4 by slightly lowering the training resolution (e.g., 480x480 instead of 512). We will clarify these trade-offs in the main paper. As suggested, we repeat the ablation of Sec. 4.3 to longer sequences (≥30s) from HDTF (test set) and TH-1KH:
> >||FVD↓| LSE-C↑|LSE-D↓|
> >|-|-|-|-|
> >| w/o self-forcing |199.08|5.177|9.082|
> >|**Ours**|**187.86**|**5.476**|**8.701**|
> >
> >The improvements in **temporal consistency (FVD)** and **lip-sync** remain consistent over longer videos. We will include this discussion in the revision.
>
> **2. Lip-sync after Stage 3**
> >We appreciate this observation and provide the requested metrics:
> >||GOHA|Portrait4D-v2|Real3DPortrait|Ours|
> >|-|:-:|:-:|:-:|:-:|
> >|**LSE-C↑**|3.070|3.384|3.401|**3.957**|
> >|**LSE-D↓**|8.322|8.404|7.967|**7.952**|
> >
> >These results confirm that **lip-sync performance is preserved after spatial adaptation**. Catastrophic forgetting is avoided through decoupled attention with modality-specific weights. **Audio parameters remain frozen during Stage 3.**
>
> **3. “Free-viewpoint” terminology**
> >Thank you for the suggestion. We will revise the wording and explicitly state the geometric limits of our method.
>
> >Our viewpoint range follows the synthetic training setup (azim ≤140°, elev ≤70°), matching typical head poses in datasets like FFHQ (see response to `WZ6a` for camera details). This corresponds to frontal-to-near-profile views, which aligns with the **talking-head scenario** and the available audio-visual training data. Back views and full 360° coverage are outside our scope. Although we do not model zoom effects, this could be done by augmenting the synthetic trajectories.
>
> >Importantly, our contribution is not to provide unrestricted 3D rendering, but to show that **view synthesis can be effectively reformulated as a conditional video generation problem**. While the current supported viewpoint range is constrained, we emphasize that the proposed formulation remains **novel and effective within the realistic conditions of talking-head generation**.
>
> **4. Lip-reading loss at high noise**
> >This loss is used after 100K steps. **Due to pre-training**, the model predicts reasonable images even from pure noise ($t=1000$) using ID+audio:
>
> >[[link]](https://drive.google.com/file/d/1c-DcMJj_f8c6EiI6PyHBuBuByaevpLAV/view?usp=sharing)
>
> >Despite blurry, **mouth shape is preserved** yielding meaningful lip features. The average lip-reading distance between such worst-case predictions and the GT video is **0.0176**. For context, the average lip-reading distance between random mismatched video pairs in our dataset is **0.0274**, indicating that even at the highest noise level, the supervision signal remains **informative rather than random**.
>
> **5. Adaptive gating**
> >We would like to clarify that, although Eq. (1) is written as a direct sum, the model already implements **an effective form of adaptive gating in practice.**
>
> >Eq. (1) reflects the **test-time case**, where all conditioning signals are simultaneously available. During training, however, the model is **not exposed to all modalities at once**. Specifically, the conditioning can be expressed as:
>
> >$$z_{out}^{cross}=Attention_{id}(Q, K, V)+\lambda_a\cdot Attention_{a}(Q,K_a,V_a)+\lambda_c\cdot Attention_{c}(Q,K_c,V_c)$$
>
> >where $\lambda_a=1,\lambda_c=0$ for Stages 1-2, and $\lambda_a=0, \lambda_c=1$ for Stage 3. Additionally, the ID attention branch is **kept frozen** throughout, explicitly preventing interference between ID and newly introduced conditioning signals.
>
> >Beyond that, we apply **independent modality dropout** ($p=0.1$) as part of classifier-free guidance. This randomly sets $\lambda_a$ or $\lambda_c$ to 0 during training, exposing the model to a range of conditioning combinations (inc. missing or partial inputs). Thus, the model learns to dynamically adjust its reliance on each signal depending on availability and context.
>
> >Taken together, the above induce an **adaptive mixing behavior at training**, while continuous $\lambda_a,\lambda_c$ scales ([0,1]) can be used at inference to adjust each signal’s influence (we use $\lambda_a=1,\lambda_c=1$ by default).
>
> >We will revise Eq. (1) and its description to clarify this.
>
> **6. User study scale**
> >We expanded the study with 22 new users (given the time constraints of the rebuttal period), but doubled the number of videos per user (30 vs. 15):
> >| |Hallo3|EchoMimic|FLOAT|Ours|
> >|:-:|:-:|:-:|:-:|:-:|
> >|**Pref.**|28.0%|21.0%|18.0%|**33.0%**|
> >
> >Our method continues to achieve the highest preference in terms of naturalness. We will further scale up the study in the revision.

---

> > ### Author Rebuttal · Reviewer_kg1S · 2026-04-03
> >
> > Thank the authors for their response. Most of my concerns have been resolved. Two minor issues remain: the user study lacks statistical significance tests, and the self‑forcing depth (F=2) does not fully validate long‑term exposure bias over very long sequences. Nevertheless, considering the rebuttal and other reviewers’ feedback, I will raise my score.

---

> > > ### Author Response · Authors · 2026-04-06
> > >
> > > Thank you for your detailed and constructive feedback. We sincerely appreciate your time and effort for reviewing our work. We are glad that our responses have clarified the key points and addressed your concerns, supporting your final positive evaluation of the work.

---

### Official Review · Reviewer_XBKC · 2026-03-11

**Soundness:** 3
**Presentation:** 3
**Significance:** 3
**Originality:** 1
**Overall Recommendation:** 5
**Confidence:** 3

**Summary:**

The authors propose a unified framework for identity-aware audio-driven portrait animation with support for novel view synthesis. They inflate an identity-aware image diffusion backbone to perform video generation with additional temporal blocks, and propose a decoupled multi-source cross-attention mechanism that aggregates attention across audio, camera, and identity streams. A reference network is also used to guide the diffusion network with reference-derived features. The framework employs autoregressive training via self-forcing, and incorporates a lip-reading network to introduce a lip synchronisation loss. Training follows a 3-stage progressive strategy: (1) audio-conditioned motion learning with ID consistency, (2) autoregressive self-forcing training with LoRA layers for reference and context conditioning, and (3) camera-conditioned view synthesis training.

**Compliance With Llm Reviewing Policy:**

Affirmed.

**Final Justification:**

The authors have clarified all of my questions. After looking at the overall rebuttals, I increase my rating.

**Key Questions For Authors:**

1. The authors only report the parameter counts for the additional transformer blocks and the original network. What about the audio encoder, camera encoder, reference network, and lip-reading network? A meaningful measure of the framework’s scale should aggregate all of these. Similarly, the computational overhead of each component warrants individual study, not just the lip-reading loss and recursive training.
2. Have the authors conducted ablations to check for information leakage? For instance, what is the model’s performance without audio conditioning? If removing audio conditioning does not significantly degrade performance, it would suggest the model is not effectively learning from the audio signal. Have the authors conducted analogous experiments for all conditioning signals (e.g., identity, camera)?
3. Have the authors tried visualising the attention maps (whether for identity, audio, or camera)? Analysing what each attention layer learns would be informative, particularly to verify that they are capturing the intended conditioning signals and not overlapping in unintended ways.
4. For novel view synthesis, is it possible to vary the camera viewpoint dynamically across frames?

### General comment
The authors propose a detailed and ambitious framework for identity-aware audio-driven portrait animation with novel view synthesis support, and it is commendable that they have managed to make it work with so many moving parts. I believe this is genuinely good work, but not well-suited for ICML. With a more thorough ablation study and submission to a more relevant venue, this work has significant potential.

**Limitations:**

Yes

**Strengths And Weaknesses:**

### Strengths
- The authors propose a detailed framework for identity-aware audio-driven portrait animation with support for novel view synthesis, successfully integrating multiple components, networks, and training strategies.
- The paper is well-written and easy to follow.
- The generated outputs look good and have been evaluated through a user study.

### Weaknesses
- Despite the paper being easy to read, the sheer number of proposed components makes it progressively difficult to understand where the performance gains are coming from, especially within the constraints of an 8-page paper.
- While the authors have clearly put significant thought into the framework and achieved strong qualitative and quantitative results, none of the individual components represent a novel contribution in isolation. The primary contribution lies in the design and integration of these components. In my view, this work would benefit from being more detailed and would be better suited for a journal submission or a venue like ACM MMSys.
- The framework has many moving parts, yet the computational and training overhead of each component is not discussed. Furthermore, the large number of components significantly expands the ablation space that ideally needs to be covered.

---

> ### Author Rebuttal · Authors · 2026-03-30
>
> We sincerely thank the reviewer for providing constructive feedback, acknowledging the strength of our results and finding our work "commendable", "genuinely good" and "ambitious". We hope our response fully resolves the concerns regarding venue suitability, as well as the technical questions.
>
> **1. Suitability for ICML**
> >We respectfully disagree that ICML is not a suitable venue. Our work focuses on core machine learning challenges, including diffusion-based generative modeling, multimodal learning, and sequence generation under autoregressive rollout. These are central ICML topics. Moreover, ICML explicitly includes **“application-driven machine learning”**, under which our work falls.
>
> >Crucially, our contribution is not purely engineering: we introduce **a new problem formulation in an active ML research area (unified spatio-temporal identity-aware generation)** and demonstrate that a carefully designed diffusion framework can solve it effectively (see also our our response to `WZ6a` about novelty).
>
> >The positive assessments by other reviewers (two **Accepts**) further support its relevance.
>
> **2. Missing details and computational overhead**
> >We thank the reviewer for the suggestion and will amend the paper to clarify any missing details as below:
>
> >**Parameter breakdown:**
> >- Reference network: 0.9B (frozen; no training overhead, shared weights at inference)
> >- Audio + camera modules: 0.2B (13% overhead)
> >- Lip-reading network: 40M (training-only, frozen)
>
> >Importantly:
> >- In contrast to **all prior works** (e.g., Wei et al., 2024; Chen et al., 2025) that train the reference network (initialized from a generic T2I backbone), **we keep it frozen**, as we found its identity-tailored weights (initialized from Arc2Face) sufficient to capture the reference appearance without further tuning. Our reference network, thus, introduces **no additional memory (shared weights with the spatial layers of the denoising UNet)**, whereas competing works inherently use different checkpoints for the two networks.
> >- Lip-reading adds **no inference cost**.
> >- Overall, our model remains **significantly more efficient than DiT baselines (10× faster)**.
>
> **3. On information leakage and ablations**
> >We provide the requested ablations for **removing audio and identity conditions in our response to** `WZ6a` (along with more ablations requested by `WZ6a`). **Please, check there for results**. Here, we additionally report the ablation for the 3D-aware NeRSemble evaluation by **removing the camera** condition:
>
> >|  | FID↓ | FVD↓ | LPIPS↓ | SSIM↑ | PSNR↑ |
> >|--------|---------|----------|--------|----------|------------|
> >| w/o camera cond.  | 35.21| 260.74 | 0.611 | 0.533 | 10.037 |
> >| **Ours** | **29.52** | **222.89** | **0.477** | **0.577** | **14.219** |
>
> >Removing any conditioning signal leads to a **consistent and significant performance drop**, confirming that the model does not rely on information leakage but instead **effectively utilizes each modality**.
>
> **4. Attention visualization**
> >In the figure below, we provide attention map visualizations from a representative cross-attention layer of the model, which confirms the capturing of intended information, particularly the localized behavior of audio features:
>
> >[[link]](https://drive.google.com/file/d/1DKT1xvwq0T9ocA8Qf1Rej3tnPB3gpxp9/view?usp=sharing)
>
> >Specifically:
> >- **Audio attention** is highly localized, focusing on speech-relevant regions (e.g., lips, eyelids).
> >- **Camera attention** is more spatially distributed, as viewpoint affects the full image (foreground and background). Some strong responses appear near facial contours, which are more sensitive to pose changes than interior face regions.
> >- **Identity attention** influences both face and background. While the background response may seem unexpected, it stems from the Arc2Face backbone, which relies solely on identity features to drive full-image generation. In addition, identity embeddings are known to inadvertently encode non-ID related information, such as accessories and mild contextual cues around the face.
>
> >*Note that attention maps are computed in the VAE's latent space (upsampled for visualization), so interpretations should be taken cautiously.*
>
> **5. Dynamic camera control**
> >Yes, our framework uses per-frame viewpoint conditioning and supports dynamic camera control across time (see examples above).

---

> > ### Author Rebuttal · Reviewer_XBKC · 2026-04-03
> >
> > Thanks to the authors for resolving all the questions. After looking at the overall rebuttals, I increase my rating. However, I highly recommend the authors to add all the ablations in the supplementary as there are multiple components proposed and described in the work.

---

> > > ### Author Response · Authors · 2026-04-06
> > >
> > > Thank you for your constructive feedback. We greatly appreciate your time and careful evaluation. We are pleased that our clarifications have addressed your concerns and will incorporate them into the revision for improved clarity. We are grateful for your positive assessment of our work.

---

### Official Review · Reviewer_WZ6a · 2026-03-13

**Soundness:** 3
**Presentation:** 4
**Significance:** 4
**Originality:** 3
**Overall Recommendation:** 5
**Confidence:** 4

**Summary:**

**Task Description**
- This paper addresses audio-driven portrait animation and free-viewpoint talking portrait synthesis, where the goal is to generate temporally coherent talking head videos from a single reference image (or identity embedding) and driving audio, optionally with continuous camera viewpoint control, without relying on explicit 3D representations such as tri-planes or NeRFs.

**Research Motivation**
- Existing 2D speech-to-video diffusion models depend heavily on reference image guidance, producing overly static sequences with limited motion diversity. Meanwhile, 3D-aware talking portrait methods rely on inversion through pre-trained tri-plane generators, which often leads to imperfect reconstructions and identity drift. This paper is motivated by the observation that a pre-trained identity-aware image diffusion backbone (Arc2Face) can be extended to video generation through lightweight architectural adaptations, enabling both audio-driven animation and implicit 3D-aware view synthesis within a unified spatio-temporal framework.

**Contributions**
- The paper introduces STARCaster, a spatio-temporal autoregressive video diffusion model built on Arc2Face (SD 1.5), extended with: (i) temporal transformer blocks for cross-frame coherence, (ii) a decoupled multi-source cross-attention mechanism for disentangled identity/audio/camera conditioning, (iii) a reference UNet for appearance injection, (iv) lip-reading supervision for improved audio-visual synchronization, and (v) a self-forcing autoregressive training strategy to address exposure bias. A three-stage progressive training recipe (audio-driven motion learning, autoregressive self-forcing, temporal-to-spatial adaptation on synthetic multi-view data) enables the unified framework to handle both tasks.

**Compliance With Llm Reviewing Policy:**

Affirmed.

**Final Justification:**

STARCaster presents a unified spatio-temporal autoregressive video diffusion model for both audio-driven portrait animation and free-viewpoint synthesis, built on Arc2Face (SD 1.5) with lightweight architectural extensions including temporal transformer blocks, decoupled multi-source cross-attention, a reference UNet, lip-reading supervision, and self-forcing training. The three-stage progressive training recipe elegantly overcomes the scarcity of 4D audio-visual data.

**Soundness.** The experimental evaluation is comprehensive: state-of-the-art FID (24.89/16.86) and FVD (185.24/145.35) on TH-1KH and Hallo3, strong NeRSemble cross-view results, a user study, and 10x faster inference than the 5B-parameter Hallo3 baseline. The rebuttal further strengthened this dimension with extended ablations demonstrating that each component and conditioning stream contributes meaningfully, and that removing any leads to clear degradation. The clarification on camera parameterization and preprocessing adds reproducibility.

**Originality.** While the individual components are adapted from several prior works, the rebuttal persuasively argues that the contribution lies in formulating a previously unaddressed unified problem and providing non-trivial adaptations, particularly the diffusion-based lip-reading supervision on noisy latents and the self-forcing adaptation for identity-aware generation. The progressive training recipe that enables learning each modality independently is a well-designed strategy. The paradigm of reformulating spatial view synthesis as conditional video generation, bypassing complex 3D inversion, is a conceptually clean contribution.

**Presentation.** The paper is clearly written and well-structured throughout. Especially, the well-represented pipeline overview (Fig. 3), qualitative comparisons, and honest discussion of computational overhead (Appendix A.3) are commendable.

**Significance.** Unifying audio-driven portrait animation and free-viewpoint synthesis in a single framework that eliminates the need for explicit 3D representations is a practically valuable contribution. The strong quantitative results across multiple benchmarks and tasks, combined with the 10x inference speedup over DiT baselines, suggest high potential for adoption. The out-of-distribution generalization to in-the-wild faces with challenging accessories further supports the practical applicability.

**Overall assessment.** The rebuttal addressed all three of my concerns satisfactorily: extended ablations validated the architectural design choices (W3), the novelty was convincingly reframed around unified problem formulation and non-trivial adaptations (W2), and additional implementation details and qualitative results addressed the domain gap concern (W1). The paper makes a solid and well-validated contribution to both audio-driven portrait animation and novel view synthesis, with a clean architectural design and strong empirical results. I maintain my score of **5 (Accept)** and positively support the acceptance of this paper.

**Key Questions For Authors:**

- (Related to W1) How well does the learned camera control generalize beyond the synthetic training distribution? Additional examples or quantitative analysis on in-the-wild faces with diverse appearance (e.g., glasses, hats, varying skin tones) would be informative.
- (Related to W1) Further clarification on implementation details would be appreciated. (i) Regarding the camera parameters, are they represented in a relative position form, or is some other normalization scheme applied? This choice is likely to have a significant impact on the generalization capability for free-view rendering. The description in Sec. 3.1 alone is insufficient to fully understand this aspect. (ii) Are there any differences in preprocessing or internal computation between the multi-view image (spatial sequence) input scenario and the video sequence input scenario? The additional details provided in Appendix A.1 are still not sufficient for a clear understanding.
- (Related to W3) Could the paper include an ablation comparing decoupled vs. single-stream cross-attention conditioning? This would help validate one of the key architectural contributions.

**Limitations:**

yes

**Strengths And Weaknesses:**

### **Strengths**
- S1. (Significance/Originality) Unifying audio-driven portrait animation and free-viewpoint synthesis in a single diffusion framework is a notable contribution that eliminates the need for a separate 3D reconstruction pipeline required by prior methods (Li et al., 2023b; Ye et al., 2024). The reformulation of view control as a video generation conditioning problem is a clean and practical design choice, and the three-stage progressive training recipe is a well-conceived strategy for overcoming the scarcity of 4D audio-visual data.
- S2. (Soundness) The decoupled multi-source cross-attention (Eq. 1), which shares queries across parallel identity/audio/camera attention streams with modality-specific key/value projections, is a principled mechanism for preserving the pre-trained Arc2Face identity prior while learning new conditioning signals independently. This design also enables flexible inference modes within a single model, which is practically valuable.
- S3. (Soundness) The self-forcing training strategy and lip-reading supervision are each validated by ablation (Table 3): removing self-forcing degrades FVD from 185.24 to 197.28 on TH-1KH, and removing lip-reading degrades LSE-D from 8.724 to 8.794. Fig. 8 provides convincing visual evidence for the lip-reading loss. The honest discussion of computational overhead for both components (Appendix A.3) adds transparency.
- S4. (Soundness/Significance) STARCaster achieves the best FID (24.89/16.86) and FVD (185.24/145.35) on TH-1KH and Hallo3, outperforming Hallo3 (a 5B-parameter DiT) with 10x faster inference (2.2 min vs. 21.4 min, Table 4). The user study (Fig. 6) further validates perceptual quality. The strong results on NeRSemble (Table 2) additionally demonstrate cross-view consistency, confirming that the unified framework works effectively for both tasks.
- S5. (Presentation) The paper is clearly written and well-structured. Fig. 3 provides an effective overview of the full pipeline, and the qualitative comparisons (Fig.s 4, 7, 8, 11) are well-selected to highlight the method's advantages in lip sync accuracy and view stability.

---

### **Weaknesses**
**Moderate**
- W1. (Soundness) The spatial training stage relies on 20K synthetic 3D heads (Li et al., 2024), and the domain gap between synthetic renderings and real faces is not explicitly quantified. The NeRSemble evaluation (Table 2) is conducted in controlled studio conditions with 16 cameras. Additional analysis of how well the learned camera control transfers to in-the-wild faces with diverse appearance would further validate the approach.
- W2. (Originality) Each major component is adapted from prior work: temporal transformers (AnimateDiff), reference UNet (MasaCtrl), self-forcing (Huang et al., 2025), lip-reading loss (Filntisis et al., 2022; Prajwal et al., 2020). The novelty lies in their thoughtful integration on Arc2Face with a progressive training recipe. While this systems-level contribution is valuable and non-trivial, the design choices are well-motivated and the resulting framework is effective, the paper could benefit from more explicitly articulating what is new beyond the integration.

**Minor**
- W3. (Soundness) Only lip-reading loss and self-forcing are ablated (Table 3). Ablations for the decoupled multi-source cross-attention and progressive training stages would further strengthen the claims about these design choices.

---

---

> ### Author Rebuttal · Authors · 2026-03-30
>
> We thank `WZ6a` for the thoughtful and supportive evaluation, and for recognizing the practical value, transparency, and strong results of our work. We address the questions below.
>
> **1. Additional ablations**
> >Following suggestions from `WZ6a` and `XBKC`, we provide extended ablations (same audio-driven setup as Tab. 3):
> >||FVD↓ (TH)|FVD↓ (Hallo3)|LSE-C↑ (TH)|LSE-C↑ (Hallo3)|LSE-D↓ (TH)|LSE-D↓ (Hallo3)|Pose Std↑ (TH)|Pose Std↑ (Hallo3)|
> >|-|:-:|:-:|:-:|:-:|:-:|:-:|:-:|:-:|
> >|w/o progr. training|191.46|152.98|5.342|6.233|8.789|8.611|4.695|4.361|
> >|w/o audio cond.|418.88|378.01|0.236|0.259|15.309|14.815|3.420|2.839|
> >|w/o identity cond.|369.29|364.02|3.105|4.003|10.631|9.292|3.761|3.101|
> >|**Ours**|**185.24**|**145.35**|**5.493**|**6.292**|**8.724**|**8.540**|**4.896**|**4.760**|
>
> >The full model performs best in all metrics.
>
> >In particular, addressing `WZ6a`’s question (W3), replacing **progressive training** (ID constraints -> reference guidance) with a single-stage autoregressive setup leads to consistent degradation, confirming its role in improving audio-visual learning.
>
> >The use of **decoupled over single-stream attention** (W3) is empirically motivated by prior portrait animation works (Chen et al., 2025; Cui et al., 2025), where separate attention is used to disentangle audio from other modalities (ID, text). Crucially, though, decoupling is **necessary for our progressive train scheme** (ID → audio-visual → spatial), where modalities are optimized independently, requiring separate weights. A naïve single-stream attention with shared weights would entangle gradients across modalities.
>
> **2. Novelty**
> >We respectfully emphasize that the novelty lies in **formulating and solving a previously unaddressed unified problem**: jointly handling **audio-driven animation, ID recontextualization, and controllable view synthesis** in a single diffusion framework.
>
> >Method-wise, while our solution might seem as an elegant combination of established ideas, we highlight the below contributions:
> >
> >- **Diffusion-based lip-reading supervision**
> >
> >    While such losses exist in GANs or 3D reconstruction, integrating them into diffusion training with noisy latents is non-trivial. Our supervision-by-denoising formulation and the associated challenges are detailed in our Supp. Mat.
> >
> >- **Self-forcing adaptation**
> >
> >   We bring the self-forcing training paradigm to ID-aware talking portrait animation, where maintaining ID under rollout is challenging.
> >
> >- **Stochastic generation**
> >
> >    Unlike baselines, our model supports diverse outputs, **enabling stochastic, ID-driven generation**; this is a key functional distinction from existing pipelines and is allowed by the strong image ID prior (Arc2Face) which, in turn, necessitates the integration of some known components towards an animation model.
>
> >Finally, we propose **a new paradigm for spatial synthesis “bypassing the complex and artifact-prone inversion”, as recognized by** `kg1S`. In alignment with `HYn1` and `kg1S`, we believe our method convinces through its integration clarity, effectiveness, and our strong results on multiple datasets.
>
> **3. Domain gap (synthetic vs real) for camera control**
> >Due to the lack of "in-the-wild" multi-view datasets, we evaluated on studio datasets (Tab. 2). Despite fixed conditions, they still cover diverse **ethnicities and facial characteristics**, validating our strong generalization across IDs. For harder real-world cases (hats, glasses), we provide **qualitative results**:
>
> >[[link]](https://drive.google.com/file/d/15jpkxabDRS3xoyaxEF77rvNh2fAOvd6p/view?usp=sharing)
>
> >showing that our method maintains stable camera control. We will add more in the revision. This robustness comes from the **appearance prior learned during video pre-training**. While tuning on synthetic data introduces slight smoothing in skin details, this has limited impact in the overall performance.
>
> **4. Additional details**
> >- **Camera parameters**
> >
> >    We follow standard conventions from 3D GAN literature (e.g., EG3D, PanoHead). During synthetic data generation, heads are centered at the origin, and cameras lie on a sphere (radius 2.7m, FOV 18.837°), facing inward. We vary azimuth and elevation and use the corresponding **cam2world extrinsics (absolute representation)** as conditioning. This setup ensures alignment with the FFHQ template used for real video training.
> >
> >- **Preprocessing**
> >
> >    Both real video and synthetic multi-view training data are aligned to the same canonical space. Real videos use standard 2D landmark alignment; synthetic renderings follow the above geometric assumptions. Aside from this, the model treats sequences in identical way. At inference, we align the input portrait and estimate its initial camera pose (via an off-the-shelf estimator as in EG3D). Target azimuth/elevation trajectories are defined via absolute matrices starting from the initial pose.
>
> >We will add these details for clarity and reproducibility.

---

> > ### Author Rebuttal · Reviewer_WZ6a · 2026-04-04
> >
> > I appreciate the authors' detailed and comprehensive rebuttal, which thoroughly addresses all of my questions and concerns. As my concerns have been fully resolved, I will keep my score at 5 (Accept).
> >
> > - **W1 (Domain gap for camera control):** The qualitative in-the-wild results and the detailed camera parameterization (EG3D conventions, absolute cam2world extrinsics, canonical-space alignment) adequately address my concerns about generalization and reproducibility.
> >
> > - **W2 (Novelty):** The rebuttal convincingly reframes the contribution as formulating a previously unaddressed unified problem, with non-trivial adaptations (diffusion-based lip-reading on noisy latents, self-forcing for ID-aware generation) that go beyond straightforward component integration.
> >
> > - **W3 (Missing ablations):** The extended ablations fully resolve this concern: removing progressive training, audio conditioning, or identity conditioning each leads to clear degradation, confirming that every component and conditioning stream contributes meaningfully.
> >
> > Overall, the paper makes a solid contribution to both audio-driven portrait animation and novel view synthesis through a clean unified framework, strong empirical results across multiple benchmarks, and a 10x inference speedup over DiT baselines. The rebuttal has reinforced my positive assessment, and I maintain my recommendation of 5 (Accept).

---

> > > ### Author Response · Authors · 2026-04-06
> > >
> > > Thank you for your thoughtful feedback and for the constructive review process. We greatly appreciate your time and insightful comments. We are pleased that our clarifications have addressed your concerns and helped reinforce your positive assessment, and we are grateful for your recognition of our work.

---

### Decision · Program_Chairs · 2026-04-30

**Decision:**

Accept (regular)

**Comment:**

This paper received mixed score initially and the rebuttal is successful. After rebuttal there becomes a consensus to accept this paper for its contribution in audio-driven portrait video generation with good performance and speed-up. Some outstanding issues (but very minor one) from reviewers are suggested to be fully resolved in the final version, like ablation, user study etc.